# Water vapor estimation based on one-year data of E-band millimeter-wave link in North China

Siming Zheng [1], Juan Huo [2,3,4], Wenbing Cai [5], Yinhui Zhang [5], Peng Li [1], Gaoyuan Zhang [2,6], Baofeng Ji [2,6] and Jiafeng Zhou [7], Congzheng Han [2,3,4]

[1]School of Electronics and Information Engineering, Nanjing University of Information Science and Technology, Nanjing 210044, China
[2]Electronics and Communication Engineering Laboratory, Key Laboratory of Middle Atmosphere and Global Environment Observation, Institute of Atmospheric Physics, Chinese Academy of Sciences, Beijing 100029, China
[3]University of Chinese Academy of Sciences, Beijing 100049, China
[4]Xianghe Observatory of Whole Atmosphere, Institute of Atmospheric Physics, Chinese Academy of Sciences, Xianghe 065400, China
[5]Beijing Institute of Tracking and Telecommunications Technology, Beijing 100094, China
[6]College of Information Engineering, Henan University of Science and Technology, Luoyang 471023, China
[7]Department of Electrical Engineering and Electronics, University of Liverpool, Liverpool L69 3GJ, UK

*Correspondence to*: Congzheng Han (c.han@mail.iap.ac.cn)

**Abstract.** The amount of water vapor in the atmosphere is very small, but its content varies greatly in different humidity areas. The change of water vapor will affect the transmission of microwave link signals, and most of the water vapor is concentrated in the lower layer, so the water vapor density can be measured by the change of the near-ground microwave link transmission signal. This study collected one-year data of the E-band millimeter-wave link in Hebei, China, and used a 20  model based on the ITU-R to estimate the water vapor density. An improved method of extracting the water vapor induced attenuation value is also introduced. It has a higher time resolution and the estimation error is lower than the previous method. In addition, this paper conducts the seasonal analysis of water vapor inversion for the first time. The monthly and seasonal evaluation index results show a high correlation between the retrieved water vapor density the actual water vapor density value measured by the local weather station. The correlation value for the whole year is up to 0.95, the root mean 25  square error is as low as 0.35 $g/m^3$, and the average relative error is as low as 5.00 %. Compared with ECMWF reanalysis, the correlation of the daily water vapor density estimation of the link has increased by 0.17, the root mean square error has been reduced by 3.14 $g/m^3$, and the mean relative error has been reduced by 34.00 %. This research shows that millimeter-wave backhaul link provides high-precision data for the measurement of water vapor density and has a positive effect on future weather forecast research.

## 1 Introduction

Water vapor content varies greatly in the atmosphere, and it is the main role of weather changes (Chen and Avissar, 1994). The evaporation and condensation of water can absorb and release latent heat, which directly affects the temperature of the

ground and the air (Held and Soden, 2000), so it plays an important role in the vertical stability of the atmosphere and the structure and evolution of the convective storm system (Weckwerth, 2000; Fabry, 2006). The direction and intensity of water vapor diffusion and transportation directly affect the regional water circulation system. For inland areas where the surface is short of water and the horizontal water exchange process is relatively weak, the diffusion and transportation of water vapor are of special significance to the regional water cycle process (Trenberth, 1999). Many weather changes and natural disasters are closely related to water vapor, which is an important physical quantity for predicting rainfall, mesoscale severe weather and global climate change (Kleespies and McMillin, 1990). Therefore, the research and detection of water vapor are very important, which is helpful to improve the accuracy of numerical weather prediction models.

The ideal requirements of the water vapor detection model are high temporal and spatial resolution, wide coverage, and accurate measurements. At present, ground stations, radiosondes and satellite systems usually cannot fully meet these requirements. The humidity measurement of the near-ground weather station is the most direct way to reflect water vapor (Gu et al., 2004), but it cannot meet the requirements of high spatial resolution because it only provides point observations. The radiosonde method is the most important way to obtain the data of the vertical distribution of humidity, and its data has high accuracy and resolution (Luo et al., 2014). Due to the limitation of equipment cost, the radiosonde is only launched about 1–4 times a day and cannot accurately monitor the temporal and spatial changes of water vapor. The coverage of satellite systems is much larger than that of general monitoring systems, but there are still limitations in accurately measuring near-ground humidity (Bevis et al., 1992). However, near-surface humidity is usually a key variable for convection. Therefore, it is necessary to develop a high-quality and near-ground water vapor density measurement technology.

In telecommunication networks, microwave backhaul links are often used as wireless connections between base station towers. The millimeter-wave backhaul link is a point-to-point line-of-sight communication link that uses the millimeter-wave as the carrier of information. Studies have shown that millimeter-waves will be affected by atmospheric factors during propagation (such as dry air and water vapour), which will cause signal attenuation. Based on this feature, Messer et al. first proposed a method for monitoring near-surface rainfall and retrieved rainfall rate using a communication link (Messer et al., 2006). After that, many studies have proved the feasibility of this method to estimate rainfall (Leijnse et al., 2007; Zinevich et al., 2009; Overeem et al., 2011; Chwala et al., 2012; Messer et al., 2012; Doumounia et al., 2014; Uijlenhoet et al., 2018; Han et al., 2019; Fencl et al., 2020; Imhoff et al., 2020; Luini et al., 2020). Similarly, water vapor will also attenuate microwave link signals. In 2009, David et al. proposed a new technology to measure atmospheric humidity using data collected by wireless systems (David et al., 2009). This technology not only can detect water vapor near the ground, but also gives estimates of water vapor density values with high temporal and spatial resolution. In 2018, Alpert et al. generated an air humidity map based on Israel's commercial microwave link data and compared it with the ERA-Interim humidity map of the European Center for Medium-Range Weather Forecast (ECMWF) for the first time. The results show that the humidity map generated from the link data is more accurate (Alpert and Rubin, 2018). Subsequently, David et al. showed in a study that when using data from multiple microwave links, the performance of humidity measurement is improved, and

demonstrated the potential of this virtual sensor network to provide a wide range of humidity field observations (David et al., 2019). In this study, we used the method of estimating the water vapor based on the ITU-R model. The method is to extract the attenuation caused by water vapor from the total attenuation (received signal level, RSL) of the millimeter-wave signal.

Then, under different pressures and temperatures in the atmosphere, use the line-by-line model provided by ITU-R to inverse the water vapor density. In order to improve the quality of the inversion of the water vapor density from the microwave links, we improved the method of extracting the water vapor attenuation value. Finally, a comparison between the link inversion results and the ECMWF reanalysis is given. The resolution of the link estimation result is 1 minute, while the ECMWF is 1 day. Moreover, the time resolution in the previous studies (David et al., 2009; Alpert and Rubin, 2018) was also higher than

5 minutes. The link length used in these studies is 2-5 km, which is 4.8 km in this paper. We used one year's E-band millimeter-wave link data to evaluate the performance of this method, and for the first time performed seasonal analysis of water vapor density retrieval. The results show that this method can provide more high-quality data for water vapor research and is conducive to the prediction of severe weather.

The rest of this article is structured as follows. Section 2 introduces the materials and methods, including the system

equipment used to build E-band millimeter-wave links, the processing of weather station data, and the introduction of methods for estimating water vapor based on the ITU-R model. Section 3 is the analysis and discussion of the experimental results. Section 4 gives the conclusions of the research.

## 2 Materials and methods

### 2.1 Data Sources

The Xianghe Atmospheric Comprehensive Observation and Test Station of the Institute of Atmospheric Physics, Chinese Academy of Sciences is located in Xianghe County, Langfang City, Hebei Province, China (39°76′ N, 117°00′ E). We set up a two-way E-band millimeter-wave transmission link, one side of the link is installed on the top of a 29 m high meteorological tower, and the other side of the link is at the roof-top of a residential building. Fig. 1(a) shows the location and length of the test link and Fig. 1(b), (c) show the transmitter and receiver of the link. The link is 4.8 km long and

operates at 73 and 83 GHz. We use Siklu's E-band radio transceivers (Siklu Carrier-Grade 1000Mbps E-Band radio, 2021) to transmission signals. The device is vertically polarized and operates at a transmission power of 7 dBm. The received signal level is recorded once every 1 minute, and the quantization resolution is 1 dB. We started to collect link data in August 2020, using network monitoring software to collect the received signal level ($RSL$). As of July 2021, the total monitoring time is one year.

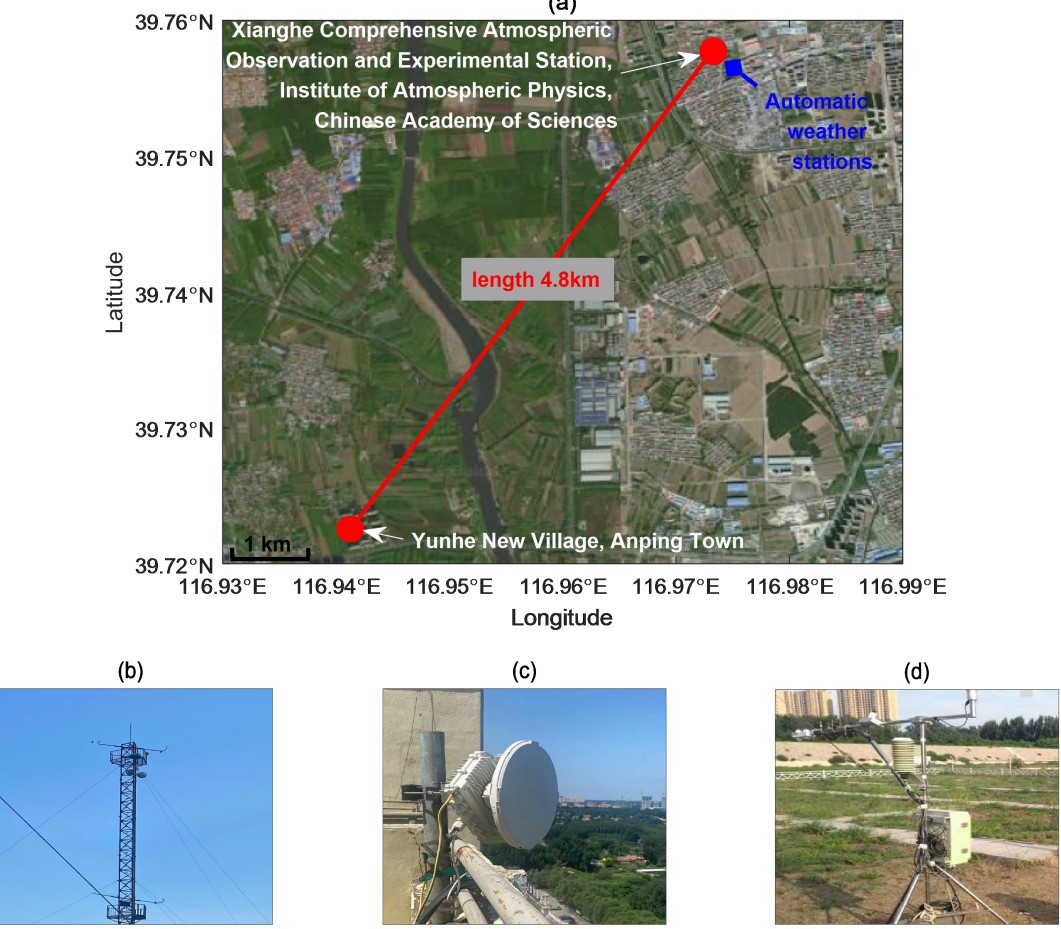

(a)

(b)    (c)    (d)

**Figure 1: (a) Location of the E-band millimeter-wave link (from Baidu Maps); (b) The transmitter of the link; (c) The receiver of the link; (d) The automatic weather stations in the experiment.**

In addition, we collected data from a weather station installed near the experimental site as a ground truth reference. Fig. 1(d) shows the automatic weather stations used in the experiment. The weather station (German OTT Parsivel$^2$ Laser Raindrop Spectrometer, 2021) is placed on the ground below the weather tower. The data is collected every minute, which is set

according to WMO standards. The data recorded by the local weather station include humidity, temperature and atmospheric pressure, where humidity is expressed in terms of relative humidity. In order to compare with the water vapor density retrieved by the link, the relative humidity of the weather station needs to be converted to the water vapor density $\rho$ (g/m$^3$) through the following formula (Liebe, 1985):

$$\rho = 1324.45 \times \frac{RH}{100\%} \times \frac{exp\left(\frac{17.67T}{T+243.5}\right)}{T+273.15}, \tag{1}$$

where $RH$ represents the relative humidity (%), and $T$ is the temperature (℃).

To more comprehensively test the link's ability to invert water vapour density values, we compare the results with the ECMWF reanalysis (CMIP5 daily data on single levels, 2021). The data source is water vapour density converted from daily near-surface relative humidity (with a horizontal resolution of 0.125° x 0.125°) obtained from ECMWF.

Due to the poor signal of the local communication base station, some data of the weather station is missing. After screening, we selected 60 dry periods with a duration of 1440 minutes per period for this experimental study and included data with a one-day (25 May 2021) duration of 1291 minutes. We excluded abnormal data from the monthly analysis. In order to make the subsequent seasonal analysis more accurate, it is ensured that each quarter contains data for 15 dry periods. Table 1 shows the data of the E-band millimeter-wave link and weather stations during the dry period each day. This includes the median values $RSL_{med}$ of the received signal level of the E-band millimeter-wave link, the median values $\rho_{med}$ of water vapor density calculated from weather station data, and the median values $p_{med}$ of atmospheric pressure and the median values $T_{med}$ of temperature measured by the weather station. These data will be used in the estimation of water vapor density. It can be seen from Table 1 that the $RSL$ varies greatly, so the use of a single attenuation baseline cannot accurately estimate the attenuation value of the water vapor density. Therefore, we consider setting a reference value for each dry period, which will be introduced in section 2.3. From the data of the weather station, it can be seen that the changes of $\rho_{med}$, $p_{med}$ and $T_{med}$ have small differences between different dry periods in the same season, and the dry periods of different seasons have large differences, so there will be seasonal differences in the estimation results of water vapor.

**Table 1: Daily variation statistics of E-band millimeter-wave link receiving signal level and weather station parameters.**

| Number | Date | | | Link 73 GHz | Link 83 GHz | Weather Station | | |
|---|---|---|---|---|---|---|---|---|
| | | | | $RSL_{med}$ (dBm) | $RSL_{med}$ (dBm) | $\rho_{med}$ (g/m$^3$) | $p_{med}$ (hPa) | $T_{med}$ (°C) |
| 1 | | | 01 | -55 | -55 | 21.90 | 1004.28 | 26.13 |
| 2 | | | 02 | -54 | -55 | 20.35 | 1000.41 | 29.19 |
| 3 | | | 11 | -59 | -60 | 20.15 | 1004.01 | 29.21 |
| 4 | | Aug | 21 | -57 | -59 | 14.93 | 1016.13 | 21.74 |
| 5 | | | 22 | -58 | -59 | 15.07 | 1011.59 | 22.49 |
| 6 | | | 27 | -58 | -59 | 19.16 | 999.22 | 26.15 |
| 7 | | | 29 | -59 | -60 | 20.39 | 1008.41 | 26.23 |
| 8 | | | 6 | -71 | -70 | 17.51 | 1008.18 | 23.89 |
| 9 | | | 7 | -71 | -70 | 16.18 | 1006.48 | 23.91 |
| 10 | | Sep | 13 | -71 | -70 | 15.02 | 1017.93 | 20.58 |
| 11 | 2020 | | 14 | -71 | -70 | 16.34 | 1014.28 | 20.53 |
| 12 | | | 16 | -70 | -69 | 6.45 | 1007.67 | 19.59 |
| 13 | | | 26 | -71 | -69 | 13.48 | 1017.84 | 16.62 |
| 14 | | | 20 | -70 | -68 | 8.58 | 1021.13 | 11.76 |
| 15 | | | 21 | -69 | -68 | 3.12 | 1017.56 | 11.16 |
| 16 | | Oct | 26 | -69 | -68 | 7.54 | 1018.19 | 12.29 |
| 17 | | | 30 | -69 | -68 | 5.61 | 1027.21 | 8.90 |
| 18 | | | 31 | -69 | -68 | 485 | 1018.85 | 9.14 |
| 19 | | | 05 | -69 | -68 | 5.68 | 1019.98 | 7.06 |
| 20 | | Nov | 14 | -69 | -68 | 6.28 | 1027.28 | 7.91 |
| 21 | | | 15 | -69 | -68 | 5.99 | 1024.98 | 4.76 |

| | | | | | | | | |
|---|---|---|---|---|---|---|---|---|
| 22 | | | 19 | -69 | -68 | 4.39 | 1015.30 | 5.08 |
| 23 | | | 06 | -68 | -67 | 2.29 | 1027.90 | 0.31 |
| 24 | | | 09 | -68 | -67 | 2.38 | 1023.88 | -3.71 |
| 25 | | | 10 | -68 | -67 | 2.43 | 1022.54 | -0.81 |
| 26 | | | 21 | -68 | -67 | 1.73 | 1027.76 | -6.94 |
| 27 | | Dec | 22 | -68 | -67 | 2.24 | 1021.97 | -4.82 |
| 28 | | | 23 | -68 | -67 | 1.26 | 1020.72 | 3.35 |
| 29 | | | 25 | -68 | -67 | 1.57 | 1022.67 | -5.30 |
| 30 | | | 26 | -68 | -67 | 2.73 | 1020.67 | -3.46 |
| 31 | | | 27 | -68 | -67 | 2.87 | 1019.26 | -2.27 |
| 32 | | Jan | 23 | -74 | -75 | 2.22 | 1024.28 | -6.88 |
| 33 | | | 24 | -75 | -75 | 2.65 | 1025.94 | -2.99 |
| 34 | | | 24 | -49 | -49 | 3.03 | 1026.99 | -0.85 |
| 35 | | Feb | 25 | -49 | -49 | 3.75 | 1026.04 | 1.32 |
| 36 | | | 26 | -49 | -49 | 3.69 | 1027.23 | 2.69 |
| 37 | | | 27 | -49 | -49 | 3.34 | 1025.05 | 5.08 |
| 38 | | | 07 | -49 | -49 | 3.35 | 1031.01 | 0.81 |
| 39 | | | 09 | -49 | -50 | 5.40 | 1024.33 | 4.70 |
| 40 | | | 10 | -49 | -50 | 6.27 | 1024.91 | 10.11 |
| 41 | | Mar | 11 | -50 | -50 | 7.13 | 1022.71 | 12.40 |
| 42 | | | 18 | -49 | -49 | 5.51 | 1024.13 | 7.50 |
| 43 | | | 23 | -49 | -49 | 3.55 | 1010.72 | 13.74 |
| 44 | | | 28 | -49 | -49 | 2.69 | 1004.67 | 14.13 |
| 45 | | | 10 | -49 | -49 | 5.13 | 1024.24 | 14.78 |
| 46 | 2021 | | 11 | -49 | -50 | 6.63 | 1022.31 | 15.21 |
| 47 | | | 12 | -49 | -50 | 7.79 | 1016.21 | 12.16 |
| 48 | | Apr | 19 | -49 | -50 | 6.71 | 1011.29 | 17.11 |
| 49 | | | 20 | -50 | -50 | 8.64 | 1015.06 | 19.66 |
| 50 | | | 21 | -49 | -50 | 8.27 | 1016.83 | 17.29 |
| 51 | | May | 24 | -49 | -49 | 4.12 | 1021.49 | 13.99 |
| 52 | | | 25 | -49 | -49 | 5.42 | 1004.93 | 22.38 |
| 53 | | | 21 | -50 | -50 | 12.18 | 1001.42 | 26.82 |
| 54 | | Jun | 22 | -50 | -51 | 12.95 | 1008.38 | 26.23 |
| 55 | | | 27 | -52 | -52 | 17.78 | 1003.67 | 27.24 |
| 56 | | | 02 | -52 | -52 | 18.14 | 1005.09 | 25.90 |
| 57 | | | 07 | -52 | -52 | 23.78 | 1004.77 | 27.10 |
| 58 | | Jul | 09 | -52 | -52 | 20.34 | 1001.52 | 27.73 |
| 59 | | | 14 | -53 | -53 | 23.42 | 1001.53 | 26.79 |
| 60 | | | 25 | -53 | -53 | 23.31 | 1003.06 | 29.83 |

Since the quantization resolution of the equipment we have used is 1 dB and the quantification resolution of the water vapor density calculated by the weather station is 0.01 g/m$^3$, the resolution of the two data is inconsistent. This is because the GUI of the wireless communication device cannot display the received signal level with higher accuracy, resulting in the link's estimated water vapor density value with a lower quantification resolution than that calculated by the weather station. Moreover, the change of water vapor is slower than the rainfall intensity (Pu et al., 2021), and the change of water vapor attenuation is also slower than the change of rain-induced attenuation. Therefore, we perform a 60-minute moving average on the link $RSL$. The purpose is to filter out the frequent fluctuations of random errors (Schleiss and Berne, 2010), to ensure that $RSL$ is consistent with the change frequency of the water vapor density of the weather station, and to improve the

accuracy of the inversion of the water vapor density. We tested different time windows and found that 60 minutes is the most appropriate. If the time window is lower than this value, the result after the moving average will not be smooth enough, and higher than this value will make the result after the moving average excessively smooth and distorted, and the hysteresis becomes obvious. It is worth noting that the time resolution of the averaged data is still 1 minute. Fig.2(a) is the comparison effect of the link received signal level $RSL$ before and after sliding on August 1, 2020, and Fig.2(b) is the $\rho$ calculated by $RH$ of the weather station. From Fig.2(a), it can be seen that the fluctuations before sliding are large, and the results after sliding are smoother, which is similar to the fluctuation frequency of the water vapor density measured by the weather station. We can also see that the change of the link signal is positively correlated with the change of the water vapor density calculated by RH of the weather station, which indicates that the E-band millimeter-wave link has the potential to retrieve water vapor.

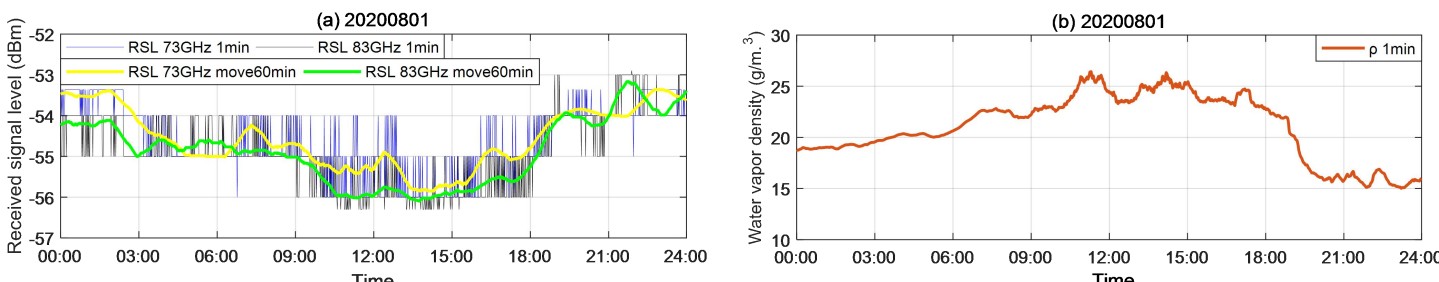

Figure 2: (a) Comparison of the link received signal level $RSL$ before and after sliding on August 1, 2020, CST; (b)Water vapor density ρ calculated from weather station data.

## 2.2 Principles of Estimating Water Vapor

Millimeter-waves are attenuated by factors such as scattering, reflection, and atmospheric absorption during transmission. As the frequency increases, the attenuation of the signal becomes larger (Uijlenhoet et al., 2018). The received signal level $RSL$ can be expressed as:

$$RSL = TSL + G_T + G_R - PL - AL - OL , \qquad (2)$$

where $TSL$ (dBm) is the transmitted signal power, $G_T$ (dBi) and $G_R$ (dBi) are the antenna gains of the transmitter and receiver, $PL$ (dB) is the propagation path loss, $AL$ (dB) is the atmospheric loss, and $OL$ (dB) for other losses. The atmospheric loss $AL$ can be expressed as follows (Daniels et al., 2014):

$$AL = A_r + A_v + A_o + A_p , \qquad (3)$$

Atmospheric loss mainly includes the attenuation effects of dry air (including oxygen), water vapor, fog and rainfall. $A_r$ (dB) is the attenuation caused by rainfall, $A_v$ (dB) is the attenuation caused by water vapor, $A_o$ (dB) is the attenuation caused by dry air, and $A_p$ (dB) is the attenuation caused by non-rainfall, such as fog, sleet and snow.

In the dry period, millimeter-waves are mainly attenuated due to the absorption of oxygen and water vapor in the lower atmosphere. This specific attenuation can be estimated using the method recommended by ITU-R P. 676-12 (Rec. ITU-R P.676-12, 2019), the formula is as follows:

$$\begin{cases} \gamma = \gamma_v + \gamma_o = 0.1820 f N^{"}(p,T,\rho,f) \\ N^{"} = \sum_i S_i F_i + N_D^{"}(f) \\ S_i = b_1 \times 10^{-1} e \theta^{3.5} exp[b_2(1-\theta)] \\ \theta = \frac{300}{T}, e = \frac{\rho T}{216.7} \\ F_i = \frac{f}{f_i} \left[ \frac{\Delta f - \delta(f_i - f)}{(f_i - f)^2 + \Delta f^2} + \frac{\Delta f - \delta(f_i + f)}{(f_i + f)^2 + \Delta f^2} \right], \delta = 0 \\ \Delta f = b_3 \times 10^{-4}(p\theta^{b_4} + b_5 e\theta^{b_6}) \\ N_D^{"}(f) = f p \theta^2 \left[ \frac{6.14 \times 10^{-5}}{d\left[1+\left(\frac{f}{d}\right)^2\right]} + \frac{1.4 \times 10^{-12} p\theta^{1.5}}{1+1.9 \times 10^{-5} f^{1.5}} \right] \\ d = 5.6 \times 10^{-4} p\theta^{0.8} \end{cases} \qquad (4)$$

$\gamma_v$: The specific attenuation due to water vapour (dB/km)

$\gamma_o$: The specific attenuation due to dry air (dB/km)

where $N^{"}$ is the imaginary part of the complex refractivity, and it is a function of the pressure $p$ (hPa), temperatures $T$ (K), frequency $f$ (GHz) and the water vapour density $\rho$ (g/m³). The $S_i$ is the strength of the i-th line (KHz), $F_i$ is the line shape factor (GHz⁻¹). $N_D^{"}(f)$ is the dry continuum due to pressure-induced nitrogen absorption and the Debye spectrum. The $e$ is the water-vapour partial pressure, $f_i$ is the line frequency and $\Delta f$ is the width of the line, $\delta$ is a correction factor which arises due to interference effects in oxygen lines and $b_1, \cdots, b_6$ are spectroscopic coefficients. In order to solve the value of $\rho$, the objective function is set as follows according to Equation (4):

$$f(x) = \frac{\gamma}{0.1820 f} - N^{"}(p,T,x,f), x = \rho, \qquad (5)$$

Therefore, solving $\rho$ is to find the roots of the nonlinear equation, that is, the value of $x$ when $f(x) = 0$.

For millimeter-wave signals at 73 GHz and 83 GHz, the specific attenuation caused by dry air is smaller than that caused by water vapor, so the specific attenuation caused by air can be ignored and the water vapor attenuation $A_v$ can be obtained:

$$A_v = \gamma \times l, \qquad (6)$$

Where $l$ (km) is the length of the link. Fig. 3 is drawn according to Equation (4), showing the relationship between the attenuation value and frequency caused by different water vapor density per 1 km for millimeter-waves in the frequency range of 0 to 100 GHz. Among them, the air pressure is 1013 hPa and the temperature is 15 ℃.

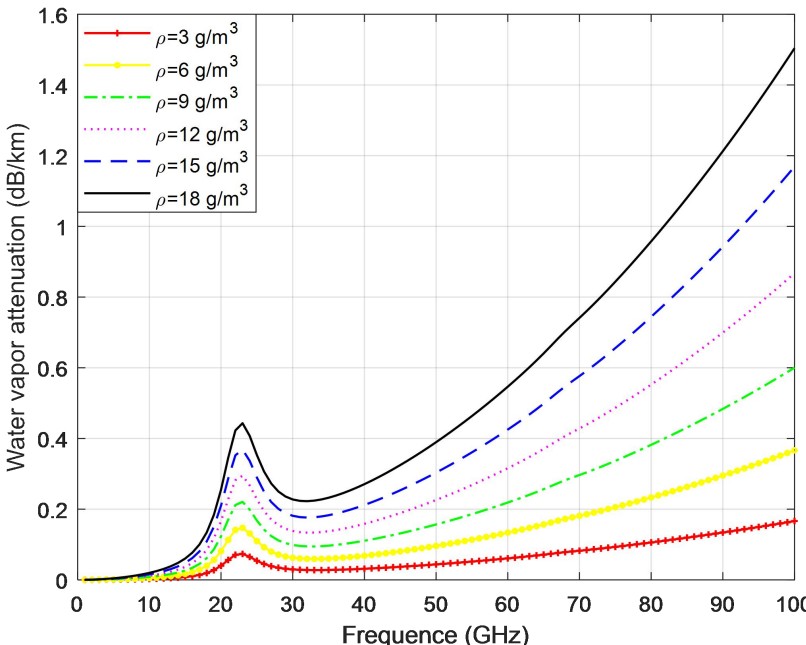

**Figure 3: When the millimeter-wave link length is 1 km, the attenuation caused by different water vapor densities in the frequency range of 0 to 100 GHz (temperature is 15°C and atmospheric pressure is 1013.25 hPa).**

Therefore, given the atmospheric temperature $T$, pressure $p$ and link frequency $f$ using the known relationship between $N''$ and $\rho$, the water vapor density $\rho$ (g/m³) can be numerically estimated by formula (4).

In order to extract the attenuation value of water vapor from all the attenuation, we set a reference value for each dry period.

During the dry period, the attenuation fluctuation of the link is mainly caused by the change of water vapor. Assuming that the received signal level is the reference value $RSL_{ref}$ when the water vapor attenuation value is zero, we obtain the reference value $RSL_{ref}$ by the following formula (6):

$$RSL_{ref} = RSL_{med} + A_{v_{med}} , \tag{7}$$

$$RSL_{med} = median(RSL_1, RSL_2, \cdots, RSL_{1440}) , \tag{8}$$

$$A_{v_{med}} = median\big(A_{v1}, A_{v2}, \cdots, A_{v1440}\big) , \tag{9}$$

where $RSL_{med}$ is the median value of the received signal level during a dry period, and $A_{v_{med}}$ is the median value of the water vapor attenuation. In order to eliminate the abnormal value caused by the influence of strong wind or equipment failure on the link, we also set the upper and lower limits of water vapor attenuation for each dry period, and determine the values of the upper and lower limits by the following formula (9):

$$\begin{cases} RSL_{low} = RSL_{med} + \big(A_{v_{med}} - A_{v_{min}}\big) \\ RSL_{up} = RSL_{med} - \big(A_{v_{max}} - A_{v_{med}}\big) \end{cases} , \tag{10}$$

$$A_{v_{min}} = min\left(A_{v1}, A_{v2}, \cdots, A_{v1440}\right),\tag{11}$$

$$A_{v_{max}} = max\left(A_{v1}, A_{v2}, \cdots, A_{v1440}\right),\tag{12}$$

Among them, $A_{v_{min}}$ is the minimum value of water vapor attenuation in a dry period, and $A_{v_{max}}$ is the maximum value, which can be obtained:

$$RSL_i = \begin{cases} RSL_{low}, & if\ RSL_i > RSL_{low} \\ RSL_{up}, & if\ RSL_i \leq RSL_{up} \end{cases},\tag{13}$$

$$i = 1,2,\cdots,1440,\tag{14}$$

Therefore, during a dry period, the attenuation value $A_{v_i}$ caused by water vapor can be determined as:

$$A_{v_i} = -RSL_i + RSL_{ref},\tag{15}$$

We collected millimeter-wave link data from August 2020 to July 2021 and selected a total of 60-day dry period data

excluding rainy period data. We have applied the above model to process the data and have retrieved the water vapor density.

## 2.3 Statistical tests

We evaluate the retrieval accuracy by calculating the Pearson correlation coefficient ($PCC$), the root mean square error ($RMSE$) and the mean relative error ($MRE$). The calculation formula is as follows:

$$PCC_k\left(X_{i,k}, Y_i\right) = \frac{1}{N-1}\sum_{i=1}^{N}\left(\frac{X_{i,k}-\mu_X}{\sigma_X}\right)\left(\frac{Y_i-\mu_Y}{\sigma_Y}\right),\tag{16}$$

$$RMSE_k = \sqrt{\frac{1}{N}\sum_{i=1}^{N}\left(X_{i,k}-Y_i\right)^2},\tag{17}$$

$$MRE_k = \frac{100\%}{N}\times\sum_{i=1}^{N}\left|\frac{X_{i,k}-Y_i}{X_{i,k}}\right|,\tag{18}$$

Among them, when $k$ is 1, $X_{i,1}$ represents the estimated water vapor density retrieved from the 73 GHz link, and when $k$ is 2, $X_{i,2}$ represents the estimated water vapor density retrieved from the 83 GHz link. $\mu_X$ and $\sigma_X$ are the average value and standard deviation of $X_{i,k}$ respectively, $Y_i$ represents the water vapor density measured by the weather station, and $\mu_Y$ and $\sigma_Y$

are the average value and standard deviation of $Y_i$ respectively. The closer the correlation coefficient is to 1, and the smaller the root mean square error and the mean relative error, it means that there is better similarity between the two data sets.

# 3 Result

## 3.1 Monthly graphs of water vapor density

We compare the estimated results of water vapor density with the actual measured values of the weather station. Fig. 4
shows the monthly summary of the water vapor density graph with a time resolution of 1 minute and presents the result
according to the season.

The results show that the water vapor inversion based on the millimeter-wave link data is positively correlated with the
observation results of the traditional weather station, and there is a good consistency, which shows that the millimeter-wave
link has great potential in estimating the water vapor density. Looking at the difference between the seasons in Fig. 4, it is
obvious that the water vapor density value is the highest in summer (June - July - August), while the water vapor density
value is the lowest in winter (December - January - February). Also, the summer months show better consistency than the
winter months. This can be explained by the climatic characteristics of Hebei, China. This area is located on the east coast of
China and belongs to the temperate humid and semi-arid continental monsoon climate. The area is hot and humid in summer
and cold and dry in winter (Climate Overview of Hebei Province, 2021). Therefore, in winter, the linear cumulative
attenuation value of water vapor on the link is too small, which makes the water vapor attenuation value unsuitable for
measurement and susceptible to noise interference (Graf et al., 2020), which is also a reason for the poor inversion results in
winter. There are many reasons for the error, and further analysis of the results is needed.

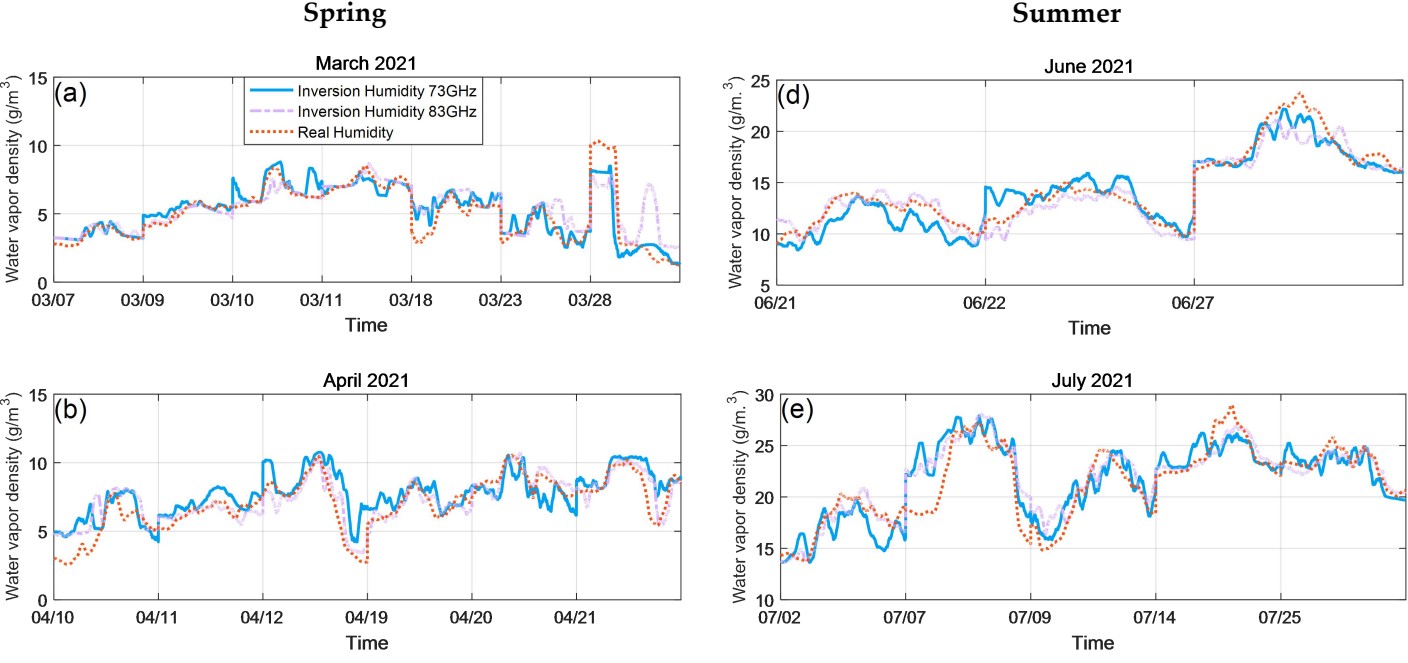

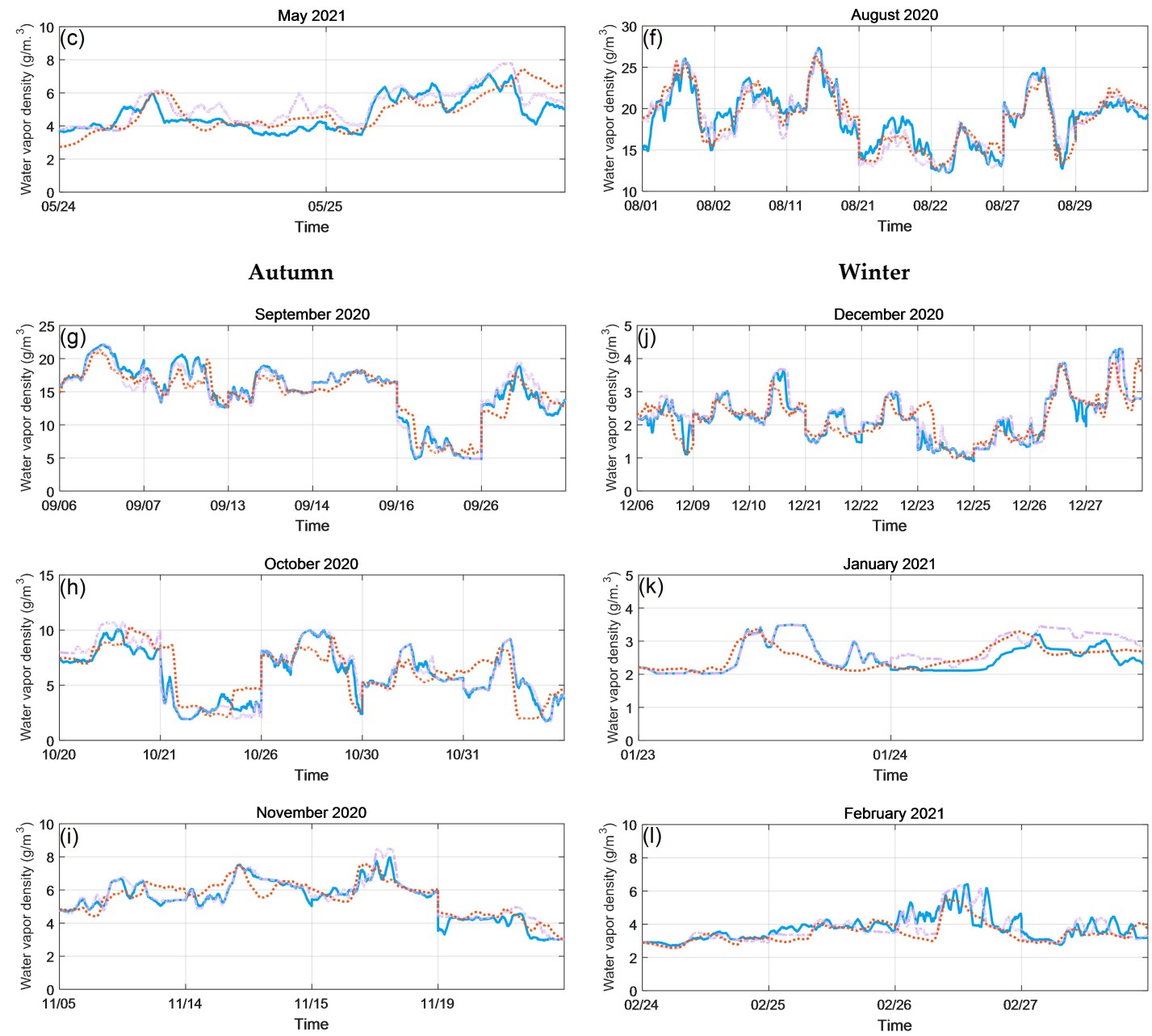

**Figure 4: The water vapor density calculated from the data obtained by millimeter-wave links on different dates is compared with the measured value of the weather station.**

**3.2 Results evaluation**

Table 2 shows the correlation, root mean square error and mean relative error between the water vapor density value retrieved by the link and the measured value of the weather station by month. In addition, we calculated the mean value of

the correlation for each quarter, the root mean square error and the mean relative error mean statistics are also summarized. It can be seen from Table 2 that the evaluation index also reflects that the the result of using millimeter-wave link to estimate

the water vapor density is good. In other words, the highest and lowest monthly correlation are 0.95 and 0.63, the highest and lowest root mean square error are 1.88 g/m³ and 0.35 g/m³, and the highest and lowest mean relative error are 27.00 % and 5.00 %. June has the highest correlation and the root mean square error and mean relative error are also low. But January, February, and May 2021 have lower correlations. Combined with Fig. 4 in Section 3.1, it can be seen that the inversion result in June is the best. In terms of seasons, the water vapour density retrieval result from the 83 GHz link during summer shows

the highest correlation with the water vapour density derived from the local weather station, and the mean relative error is the lowest. Similarly, the evaluation result was better at 83 GHz in June, which shows that the 83 GHz millimeter-wave link has greater potential for water vapor inversion.

**Table 2: Correlation, root mean square error and mean relative error between the water vapor density obtained by millimeter-wave links in different months and the measured value of the weather station, as well as the average value of each quarter.**

| | | 2021 | | | | | | | 2020 | | | 2021 | |
| | | Spring | | | Summer | | | | Autumn | | | Winter | |
| | | Mar | Apr | May | Jun | Jul | Aug | Sep | Oct | Nov | Dec | Jan | Feb |
| 73 GHz | $PCC_1$ (-) | 0.89 | 0.79 | 0.64 | 0.94 | 0.87 | 0.91 | 0.94 | 0.79 | 0.87 | 0.83 | 0.63 | 0.69 |
| | $PCC_{avg}$ (-) | | 0.77 | | | 0.91 | | | 0.87 | | | 0.72 | |
| | $RMSE_1$ (g/m³) | 0.91 | 1.34 | 0.89 | 1.29 | 1.88 | 1.45 | 1.57 | 1.48 | 0.57 | 0.39 | 0.35 | 0.61 |
| | $RMSE_{avg}$ (g/m³) | | 1.05 | | | 1.54 | | | 1.21 | | | 0.45 | |
| | $MRE_1$ (%) | 13.00 | 19.00 | 14.00 | 8.00 | 7.00 | 6.00 | 9.00 | 24.00 | 8.00 | 13.00 | 10.00 | 12.00 |
| | $MRE_{avg}$ (%) | | 15.33 | | | 7.00 | | | 13.67 | | | 11.67 | |
| 83 GHz | $PCC_2$ (-) | 0.83 | 0.88 | 0.78 | 0.95 | 0.90 | 0.94 | 0.93 | 0.78 | 0.84 | 0.81 | 0.71 | 0.85 |
| | $PCC_{avg}$ (-) | | 0.83 | | | 0.93 | | | 0.85 | | | 0.79 | |
| | $RMSE_2$ (g/m³) | 1.18 | 0.92 | 0.81 | 1.17 | 1.66 | 1.19 | 1.59 | 1.63 | 0.62 | 0.42 | 0.40 | 0.47 |
| | $RMSE_{avg}$ (g/m³) | | 0.97 | | | 1.34 | | | 1.28 | | | 0.43 | |
| | $MRE_2$ (%) | 21.00 | 12.00 | 14.00 | 6.00 | 6.00 | 5.00 | 9.00 | 27.00 | 9.00 | 14.00 | 12.00 | 9.00 |
| | $MRE_{avg}$ (%) | | 15.67 | | | 5.67 | | | 15.00 | | | 11.67 | |

Fig. 5 shows more clearly the value of daily and monthly evaluation indexes. It can be seen from Fig. 5 that the correlation and mean relative error in summer performed well, but the root mean square error performed poorly. In contrast, the root mean square error in winter is lower. This is because the water vapor density in winter is low, so the error is relatively low, while the water vapor density in summer is high, so the error is relatively large. The estimation errors in spring (March - April - May) and autumn (September - October - November) still exist, but they are relatively low compared to winter.

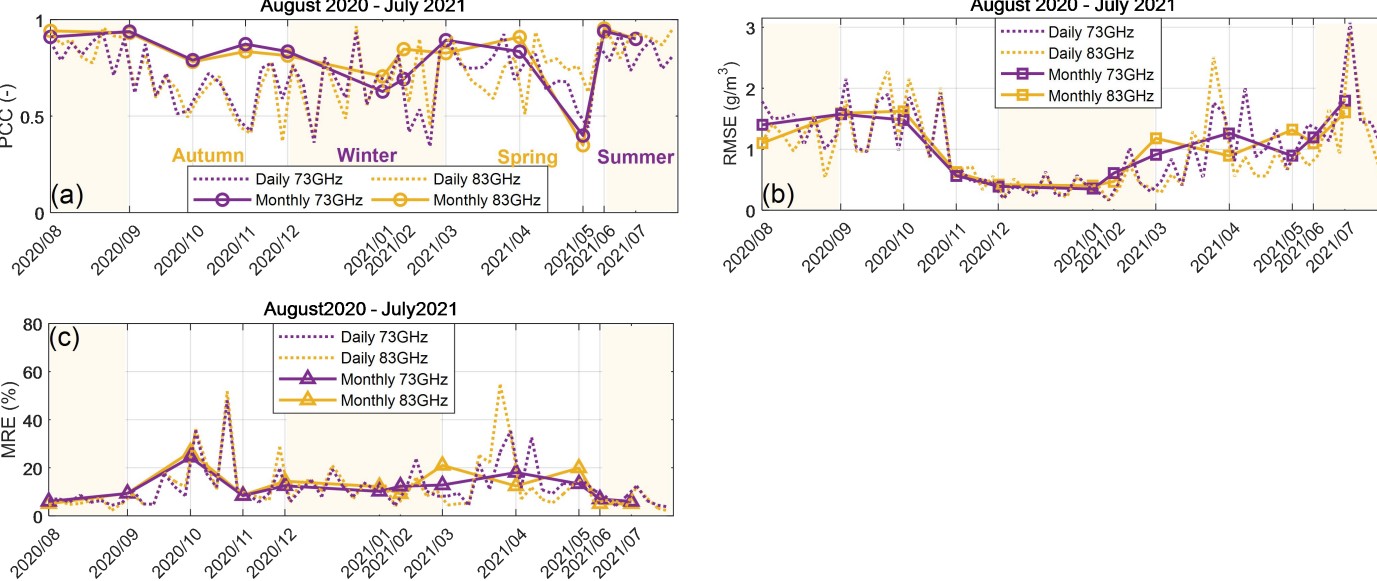

Figure 5: Daily and monthly evaluation index of water vapor density inversion(a) the correlation between the derived water vapor density and the reference, and (b) the root mean square error between the derived water vapor density and the reference. (c) the mean relative error between the derived water vapor density and the reference.

### 3.3 Comparison of daily water vapor density from links, ECMWF, and weather station data

Fig. 6 shows the daily water vapor density during August 2020 to July 2021 including link result, ECMWF reanalysis (CMIP5 daily data on single levels) versus the weather station measurement. The daily near surface relative humidity obtained from ECMWF with a horizontal resolution of 0.125° x 0.125° is converted to water vapor density. The estimated result of the link and the actual measurement of the weather station have been averaged per day, which is more convenient to compare with the result of ECMWF. From the perspective of daily water vapor density changes, the estimated results of the millimeter-wave link are closer to the actual measurements of the weather station, and the correlation reaches 0.99, while the ECMWF forecast results is worse, that is, the correlation is 0.82.

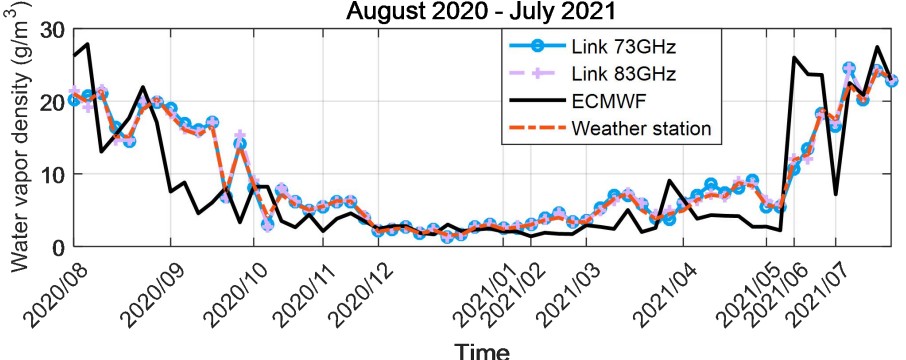

$PCC_{Link_{73}-WS}$ : 0.99
$PCC_{Link_{83}-WS}$ : 0.99
$PCC_{ECMWF-WS}$ : 0.82
$RMSE_{Link_{73}-WS}$ : 0.65 g/m³
$RMSE_{Link_{83}-WS}$ : 0.58 g/m³
$RMSE_{ECMWF-WS}$ : 4.79 g/m³
$MRE_{Link_{73}-WS}$ : 6.00 %
$MRE_{Link_{83}-WS}$ : 5.00 %
$MRE_{ECMWF-WS}$ : 39.00 %

**Figure 6: The water vapor density graphs from the link, the ECMWF reanalysis and the weather station measurement. These include the correlation, root mean square error and the mean relative error between the link and the weather station measurement, the ECMWF and the weather station measurement. WS = weather station.**

## 4 Discussion

Since the measurement from the millimeter-wave link is a linear accumulation of integrated data, while the traditional weather station provides point measurements. Therefore, the estimated result of the link estimation will be different from the data of the weather station. In addition, the weather station in this paper is placed at one side of the link and the observation from a weather station is only representative of the very local area where the equipment is sited which may cause some deviations between the estimated results and the actual measured results. There are also some environmental factors, such as the influence of wind speed, fog and dew on the antenna. This will reduce the accuracy of link inversion of water vapor density. As shown in Fig. 4, the results of the link inversion have the same trend as the measured values of the weather station, but they are not always consistent.

There are some unusual results in the measurement of the link. For example, from March 28 in Fig. 4 (a), the water vapor density estimated by the 83 GHz link is significantly higher than the estimated result of the 73 GHz link and the actual value measured by the weather station. This may be related to the extraction of attenuation. For this dry period, there is only one baseline that may not be able to accurately extract the attenuation caused by water vapor, and a more real-time reference value is needed. Although the calculation of the baseline may be one of the reasons for the error, this research has improved the real-time performance of the baseline and reduced errors compared with the studies of David et al., Alpert et al., and Fencl et al. (only set a constant baseline) (David et al., 2009; Alpert and Rubin, 2018; Fencl et al., 2020). Also, in Fig. 4 (c) and 4 (h), the estimated result of the water vapor density of the link decreases at the night of the day, which is caused by other uncertain sources. For example, changes in atmospheric refractive index and hardware-related artifacts may cause millimeter-wave ray bending and $RSL$ changes. This may also explain why the $MRE$ in October in Fig. 5 (c) is higher than in other months. At present, there are very few studies on seasonal analysis of water vapor inversion, and Pu et al. only conducted tests in late summer and early winter (Pu et al., 2021). The research in this paper provides seasonal analysis and high-resolution data for the inversion of water vapor density in the Xianghe area, which is expected to provide insightful information for weather monitoring in North China. The research results show that it is feasible to invert water vapor using millimeter-wave links, and this method can be extended to the monitoring of meteorological factors such as rain, snow, and fog. At the same time, it also provides a basis for atmospheric monitoring of commercial microwave links, which will help to promote applications in the field of meteorology in the future. This is a test link, but E-band links are expected to be widely used in 5G networks and smart city networking.

Moreover, the link performs better than ECMWF reanalysis in estimating water vapor density. Compared with ECMWF 's prediction results, the correlation of the daily water vapor density estimation of the link has increased by 0.17, the root mean

square error has been reduced by 3.14 g/m$^3$, and the mean relative error has been reduced by 34.00 %. It can also be seen from Fig. 6 that the predicted result of ECMWF is closer to the measured value only around winter. In fact, these three data sets work on different spatial scales, which must have a great impact on water vapor inversion results.

## 5 Conclusions

Research on the water vapor retrieval of millimeter-wave links may improve the ability to deal with extreme weather-related hazards. For example, flash floods are usually triggered by heavy rainfall. However, the "fuel" for the formation of convective rain cells that lead to such rainfall is water vapor, so more accurate measurement means better response to the dangers facing humans and their environment (Fencl et al., 2020; Harel et al., 2015). Moreover, the millimeter-wave link also has the great potential to become a water vapor monitoring sensor, which can provide higher spatial density water vapor data. We demonstrated the processing of millimeter-wave link data with a time resolution of 1 minute for 60 dry periods from August 2020 to July 2021, and used the line-by-line calculation of gaseous attenuation model provided by ITU-R to retrieve water vapor. We have proposed a new method which is to set a reference value for each dry period and give the upper and lower limits of water vapor attenuation. Then we applied this method to one year's data. We found that the water vapor density value estimated from the millimeter-wave link is highly correlated with the actual measurement value of the weather station. We performed a seasonal analysis of the results. The highest Pearson correlation coefficient in summer months is 0.95, and the average value is 0.92. The lowest mean relative error is 5.00 %, and the average value is 6.00 %. The monthly and seasonal evaluation indicators show good results. Compared with other studies, our water vapor inversion results have a higher time resolution, this is, our resolution is 1 minute, while it is 1 day for the tested ECMWF product. Moreover, the time resolution in the previous studies (David et al., 2009; Alpert and Rubin, 2018; Fencl et al., 2020; Pu et al., 2021) was equal or greater than 5 minutes. Future research can use high-resolution humidity fields to improve weather forecasts, and its significance also includes the ability to study extreme events that are mainly controlled by humidity fields.

In addition, the millimeter-wave link we use is longer, and the linear cumulative attenuation value of water vapor on the link increases, which is conducive to the measurement of water vapor density. However, the influence of free space loss and channel noise will also increase, which poses a higher challenge to the sensitivity of signal detection at the receiving end. Secondly, the microwave link is also very sensitive to mechanical oscillations. Strong winds may cause the link transmitter or receiver to move and may also interfere with the accuracy of the measurement. Therefore, this increases the difficulty of retrieving water vapor. The seasonal evaluation index shows that the millimeter-wave link has the best water vapor retrieval effect in summer, but the worst in winter. This seasonal difference is also difficult to overcome. In the future, we will consider improving water vapor monitoring in winter in our research. As the season changes, the ambient temperature also changes. We can try adding a temperature variable to the process of estimating the water vapor density.

**Author Contributions:** Experiment design, G.Z., B.J., J.Z.; instrument, B.J., G.Z., W.C., Y.Z.; measurement data collection, C.H., G.Z., B.J., J.Z., P.L.; feasibility study, C.H., B.J., J.Z., G.Z.; conceptualization, S.Z. and C.H.; methodology, S.Z.; software, S.Z.; validation, S.Z., C.H. and P.L.; formal analysis, C.H.; investigation, S.Z.; resources, C.H., G.Z., and B.J., J.Z.; data curation, J.H; writing—original draft preparation, S.Z.; writing—review and editing, S.Z., C.H., J.Z., J.H., P.L., W.C., Y.Z., G.Z., B.J.; visualization, S.Z.; supervision, P.L.; project administration, W.C. and Y.Z.; funding acquisition, C.H., W.C., Y.Z. All authors have read and agreed to the published version of the manuscript.

**Funding:** This work was financially supported in part by the National Natural Science Foundation of China (Grant No. 42027803, 41605122, 41775032, 61701172, 61801170); LAGEO of Institute of Atmospheric Physics, Chinese Academy of Sciences (LAGEO-2019-2, LAGEO-2018-1); Young Backbone Teachers in Henan Province (2018GGJS049); Henan Province Young Talent Lift Project (2020HYTP009); Program for Science & Technology Innovation Talents in the University of Henan Province (20HASTIT022). China Postdoctoral Science Foundation (2018M633351), Suzhou Qiu Shi Technology Co., Ltd.

**Institutional Review Board Statement:** Not applicable.

**Informed Consent Statement:** Not applicable.

**Data Availability Statement:** The data presented in this study are available on request from the corresponding author. The data are not publicly available due to restrictions privacy.

**Acknowledgments:** The author would like to thank Yoav Rubin for providing us with the help of link data analysis, Daniel Ephraty from Siklu for tackling the technical issues, as well as Weidong Nan, Qing Yao, Guowei An, Tiefeng Chen and Qiuzhen Yu for helping with the field measurement and maintainance. The authors thanks anonymous reviewers for providing helpful advice.

**Conflicts of Interest:** The authors declare no conflict of interest.

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
