# Peer review of "Water vapor estimation based on one-year data of E-band millimeter-wave link in North China"

_Atmospheric Measurement Techniques, 2021_

## Referee Comment (RC1)

Review of: **Water vapor estimation based on one-year data of E-band millimeter-wave link in the northeast of China** by Siming Zheng et al.

Ruben Imhoff

Ruben.Imhoff@deltares.nl

November 15, 2021

**Summary**

The authors present a case study where they use high-frequency microwave link attenuation data for water vapor density estimations using one microwave link in Hebei, China. The study is based on one-year of 1-min E-band millimeter-wave link data and the resulting water vapor density estimations are compared to observations from a weather station placed at the transmitter of the link setup. Using the equations from the ITU-R technical note, the authors manage to produce water vapor density estimations from the link attenuation that resemble the observations well, and that on a relatively high temporal resolution. I would like to thank the authors for a clear and straightforward study and a concise manuscript, which gives scientifically relevant outcomes, in my opinion.

Despite my enthusiasm about this study, I also have quite some (sometimes substantial) points for improvement and questions. What I think is currently lacking in the manuscript, is twofold:

1)  The authors use the ITU-R equations for water vapor density estimations using link attenuation as input. This is, however, an application of that method and not a new method. That is on itself not a problem, but the authors claim to have improved the model, while the improvements are not clearly stated in the manuscript. I think it would be valuable to clarify this.
2)  A discussion section in which the results of this study are benchmarked, related to existing literature and where possible pitfalls and improvements can be further explained.

Below you can find an extension of my general comments, followed by specific comments (line by line) and some technical corrections.

**General comments**

*Water vapor density estimation procedure*

The water vapor density estimation procedure is based on the ITU-R technical note. The authors mention already in the abstract that they have based their estimation procedure on the equations in this technical note and that they have used an improved method of extracting the water vapor induced attenuation value. To me, it was not directly clear at which point the authors introduce their improvements on the method(s). Can I ask the authors to clarify this in the manuscript? I.e., did the authors only use an existing method and apply this for a test case or did they improve the method (both are scientifically relevant, but will lead to different statements in abstract and conclusions).

In addition, the ITU-R recommended method is used in this work and written down in equation 4 (based on eq. 1 in the reference). Only equation 4 is, however, not sufficient to solve the equation. For that, the set of equations (eq. 1 – 9 in the ITU-R technical note) are needed. As this method is so dependent on these equations, I think it would be valuable to state them in the manuscript as well, including some explanations and correct references to the ITU-R equations. Finally, the end goal is to estimate the water vapor density $\rho$ when the attenuation is known. Although this indeed follows from the set of equations, it would be useful to add a final equation showing how $\rho$ is computed from the other terms.

*Model benchmarking*

What I have missed, is a benchmark for comparison with the estimation results. The authors regularly talk about good model results, but good compared to what? I am not sure if this will be outside the scope of this paper, but a comparison with some other estimation techniques would be very relevant information and could strengthen the statements made in the paper. For instance, how do the estimation results relate to model estimates from e.g. ECMWF forecasts or local NWP forecasts, etc.?

Besides, the least the authors probably can do, is to discuss the relationship to and expected results compared to other microwave link attenuation-based water vapor estimation techniques, such as those introduced in: Alpert and Rubin (2018), David et al. (2009, 2019), Pu et al. (2021) and Fencl et al. (2021).

*Discussions section*

In line with my previous comment about missing some discussion about the relationship to other literature and estimation techniques, I miss a discussion section in this work. It would be very insightful to get more information on reasons for any estimation-observation discrepancies we see. As mentioned before, how do the results relate to other literature? And, what is the authors' view on possible improvements?

**Specific comments**

Lines 50 – 55: Here the authors introduce the millimeter wave / commercial microwave links. Can I ask the authors to briefly introduce what they are, what they are used for and how atmospheric processes can attenuate the signal? This might be relevant introductory information for a more general audience.

Line 53 – references: This may require some extra references, e.g. (but absolutely not a complete list – but that is probably also not necessary): Messer et al. (2006), Leijnse et al. (2007), Zinevich (2009), Overeem et al. (2011), Chwala et al. (2012), Doumounia et al. (2014), Uijlenhoet et al. (2018), Fencl et al. (2020).

Lines 62 – 63 "In this study, we used the method of estimating water vapor based on the ITU-R model": Could the authors briefly introduce this method here (just in words)? The authors further elaborate on the method in the methods section, but it may be helpful for readers of the paper to get a first idea of what this model is and does.

Lines 64 – 65 "Compared with the previous method, the time resolution of the retrieved water vapor density value is improved, and the estimation error is reduced.": What was the time resolution and estimation error in this ('the previous') method, and what was the length of the link path?

Lines 78 – 80: What kind of microwave link was used, i.e. commercial or for research, which brand, etc.?

Lines 85 – 90: Could the authors provide a little more information about the weather station? On what elevation was it placed, also relative to the microwave link, was the measurement setup according to WMO standards, with what frequency is data collected? How representative are the measurements for the link path? The link path crosses a river, which may result in a slightly different humidity than what is measured at the weather station at the (seemingly) more urban site (but do correct me if I am wrong  - I would like to see the authors' thoughts on this).

Lines 97 – 98 "median values": A question more out of interest, would it be helpful to also use other statistics of the received signal or even the full signal to get an idea of the uncertainty in the estimation process?

Lines 107 – 108 "Since the quantization resolution of the equipment we have used is 1 dB and the quantification resolution of the water vapor density calculated by the weather station is 0.01 g/m, the resolution of the two data is inconsistent.": Could the authors elaborate a bit on this, also taking into account the length of the link path?

Lines 110 – 113: The moving average makes sense, I think. Have the authors, however, tested other moving window averages? I.e., where is the optimum and can we even go to higher temporal resolutions?

Lines 176 – 178 "We collected […] water vapor density.": This belongs in the methods section.

Lines 190 – 191 "There are many reasons for the error, and further analysis of the results is needed.":  Can the authors elaborate in the discussion section on this? What errors are you thinking of and is there a threshold value for the cumulative attenuation value / water vapor density that makes it unfeasible for measurement?

Lines 197 – 205 / Eq. 15 – 17: This belongs in the methods section.

Lines 209 – 211: Can I ask the authors to quantify the results in the text a bit more? Besides, what is 'good'? This needs a benchmark, so is it better than another model or estimation method?

Lines 213 – 215 "Similarly, the evaluation result was better at 83 GHz in June, which shows that the 83 GHz millimeter-wave link has greater potential for water vapor inversion.": That makes sense, indeed, seeing Fig. 2 in the manuscript.

Lines 222 – 224 "Since the […] point measurement.": What do the authors mean with this sentence?

Lines 224 – 225 "In addition, the weather station in this paper is placed at one side of the link, so the estimated result will be different from the data of the weather station.": It may be worth mentioning this in either the methods or discussion section. Plus, what would this mean for the measurements?

Lines 234 – 235 "vapor, so more accurate measurement means better response to the dangers facing humans and their environment (Fencl et al., 2020; Harel et al., 2015).": Very true. I think the authors can even make this stronger by mentioning the potential of new, opportunistic sensors here, which potentially gives a high(er) density of water vapor density sensors.

Line 236 "1 minute": But 60-min aggregations were used and validated, right?

Lines 236 – 238 "and used the […] within this year.": State here what the method was and especially what the authors did change, so what is new.

Lines 240 – 241: Besides the best reached values, also mention the average values to give the reader an idea of the overall model quality.

Lines 241 – 242 "Compared with previous studies, our water vapor inversion results have a higher time resolution.": How much higher?

Lines 250 – 251 "This seasonal difference is also difficult to overcome. In the future, we will consider improving water vapor monitoring in winter in our research.": Could the authors share any thoughts on how to do this?

Table 1: Could the authors add the number of minutes on record per indicated day? In addition, can the authors say something about the variation in the measurements during the days?

Figure 3: the colors of the figure are not color-blind proof. I would recommend using different colors or to use a set of dashes in the lines. Besides, the figure is quite similar to Fig. 2 in Pu et al. (2021).

Figure 4: What I miss in the results section (or accompanying discussion section) is a little discussion about any discrepancies we see between observations and model (e.g., but note that it is not limited to this, the differences we observe on the right of Fig. 4c and the drop in the simulations, but not the observations, in Fig. 4h). Can I ask the authors to comment on this? If the comments have a more speculative nature, this would very well fit in a discussion section. In any case, it could provide valuable information, also for the future development of such estimation methods.

Table 2 & Figure 5: The monthly overview is clear and interesting. Besides this overview, I think it may also be valuable to show a figure with at least some of the metrics per time step, to visualize the hour-to-hour or day-to-day discrepancies that can take place. That is valuable information for possible operational implementations of this method.

Figure 5: What causes the high MRE peak in October, compared to the other months?

**Technical corrections**

Line 30 - "The evaporation […]": The evaporation of water

Lines 44 – 45 "the radiosonde is only launched about 2–4 times a day": Often even only once a day.

Line 100 "water vapor inversion": From a consistency perspective, would it better to use water vapor density instead of inversion?

Line 170 / Eq. 12: Is this a conditional statement? If yes, the notation should be slightly different (to make that clear).

Line 179 "map": I think this should be 'graph'.

Line 205 "which shows that the use of millimeter-wave link signal attenuation can estimate the water vapor density very well.": I think this can be removed, as the rest of the sentence already explains enough.

Line 239 "obtained": I think this should be 'estimated'.

Line 240 "PCC" and "MRE": Better to write that out here, so Pearson's correlation coefficient and mean relative error.

Figure 4: Something seems to go wrong with the figure headings, 'Spring and Summer' are on a different page than the rest of the figure. It is possible to either make the figure smaller or to make two figures out of it (e.g. spring + summer and autumn + winter). Besides, for comparison purposes, it helps when the y-axes are the same throughout the figure.

**References**

Alpert, P. & Rubin, Y. (2018). First Daily Mapping of Surface Moisture from Cellular Network Data and Comparison with Both Observations/ECMWF Product. Geophysical Research Letters, 45(16), 8619–8628. https://doi.org/10.1029/2018GL078661

Chwala, C., Gmeiner, A., Qiu,W., Hipp, S., Nienaber, D., Siart, U., et al. (2012). Precipitation observation using microwave backhaul links in the alpine and pre-alpine region of Southern Germany. Hydrology and Earth System Sciences, 16(8), 2647–2661. https://doi.org/10.5194/hess-16-2647-2012

David, N., Alpert, P., & Messer, H. (2009). Technical Note: Novel method for water vapour monitoring using wireless communication networks measurements. Atmospheric Chemistry and Physics, 9(7). https://doi.org/2413–2418.10.5194/acp-9-2413-2009

David, N., Sendik, O., Rubin, Y., Messer, H., Gao, H.O., Rostkier-Edelstein, D., & Alpert, P. (2019). Analyzing the ability to reconstruct the moisture field using commercial microwave network data. Atmospheric Research, 219(0169-8095), 213–222. https://doi.org/10.1016/j.atmosres.2018.12.025

Doumounia, A., Gosset, M., Cazenave, F., Kacou, M., & Zougmore, F. (2014). Rainfall monitoring based on microwave links from cellular telecommunication networks: First results from a West African test bed. Geophysical Research Letters, 41, 6016–6022. https://doi.org/10.1002/2014GL060724

Fencl, M., Dohnal, M., Valtr, P., Grabner, M., & Bares, V. (2020). Atmospheric observations with E-band microwave links—Challenges and opportunities. Atmospheric Measurement Techniques, 13, 2020, 6559–6578. https://doi.org/10.5194/amt-13-6559-2020

Fencl, M., Dohnal, M., & Bares, V. (2021). Retrieving Water Vapor From an E-Band Microwave Link With an Empirical Model Not Requiring In Situ Calibration. Earth and Space Science, 8(11), e2021EA001911, https://doi.org/10.1029/2021EA001911

Leijnse, H., Uijlenhoet, R., & Stricker, J. N. M. (2007). Rainfall measurement using radio links from cellular communication networks. Water Resources Research, 43, W03201. https://doi.org/10.1029/2006WR005631

Messer, H., Zinevich, A., & Alpert, P. (2006). Environmental monitoring by wireless communication networks. Science, 312(5774), 713–713. https://doi.org/10.1126/science.1120034

Overeem, A., Leijnse, H., & Uijlenhoet, R. (2011). Measuring urban rainfall using microwave links from commercial cellular communication networks. Water Resources Research, 47, W12505. https://doi.org/10.1029/2010WR010350

Pu, K., Liu, X., Liu L. & Gao, T. (2021). Water Vapor Retrieval Using Commercial Microwave Links Based on the LSTM Network. IEEE Journal of Selected Topics in Applied Earth Observations and Remote Sensing, 14(1939-1404), 4330-4338. https://doi.org/10.1109/JSTARS.2021.3073013

Uijlenhoet, R., Overeem, A., & Leijnse, H. (2018). Opportunistic remote sensing of rainfall using microwave links from cellular communication networks. WIREs Water, 5(4), e1289. https://doi.org/10.1002/wat2.1289

Zinevich, A., Messer, H., & Alpert, P. (2009). Frontal rainfall observation by a commercial microwave communication network. Journal of Applied Meteorology and Climatology, 48(7), 1317–1334. https://doi.org/10.1175/2008JAMC2014.1

---

## Author Comment (AC1)

Dear reviewer:

Thank you for your precious comments and advice. Those comments are all valuable and very helpful for revising and improving our paper, as well as the important guiding significance to our research. We have studied comments carefully and have made correction which we hope meet with approval. Revised portion are marked in red in the paper. The main corrections in the paper and the responds to your comments are as follows:

**Specific comments:**

1.Lines 50–55: Here the authors introduce the millimeter wave/commercial microwave links. Can I ask the authors to briefly introduce what they are, what they are used for and how atmospheric processes can attenuate the signal? This might be relevant introductory information for a more general audience.

**Response:** Thank you for the comment. We have added an introduction to the millimeter wave links in the paper as follows: "In telecommunication networks, microwave backhaul links are often used as wireless connections between base station towers. The millimeter-wave backhaul link is a point-to-point line-of-sight communication link that uses the millimeter-wave as the carrier of information. Studies have shown that millimeter-waves will be affected by atmospheric factors during propagation (such as dry air and water vapour), which will cause signal attenuation." (Page 2 line 50–53).

2.Line 53－references: This may require some extra references, e.g. (but absolutely not a complete list－but that is probably also not necessary): Messer et al. (2006), Leijnse et al. (2007), Zinevich (2009), Overeem et al. (2011), Chwala et al. (2012), Doumounia et al. (2014), Uijlenhoet et al. (2018), Fencl et al. (2020).

**Response:** We agree with the comment and quoted these articles in the paper (Page 2 line 55-57).

3.Lines 62–63 "In this study, we used the method of estimating water vapor based on the ITU-R model": Could the authors briefly introduce this method here (just in words)? The authors further elaborate on the method in the methods section, but it may be helpful for readers of the paper to get a first idea of what this model is and does.

**Response:** We agree with the comment. We have added a brief introduction to this method in the paper as follows: "In this study, we used the method of estimating the water vapor based on the ITU-R model. The method is to extract the attenuation caused by water vapor from the total attenuation (received signal level, RSL) of the millimeter wave signal. Then, under different pressures and temperatures in the atmosphere, use the line-by-line model provided by ITU-R to inverse the water vapor density." (Page 2-3 line 66–69).

4.Lines 64–65 "Compared with the previous method, the time resolution of the retrieved water vapor density value is improved, and the estimation error is reduced.": What was the time resolution and estimation error in this ('the previous') method, and what was the length of the link path?

**Response:** Thanks for your comment. We have added a comparison with ECMWF reanalysis in the result section of the paper. It is also compared with the results of these previous studies (David et al., 2009; Alpert and Rubin, 2018). We've re-written the sentence in the manuscript as the following: "Compared with ECMWF reanalysis, the time resolution of the retrieved water vapor density value is improved, and the estimation error is reduced. The resolution of the link estimation result is 1 minute, while the ECMWF is 1 day. The correlation between the link estimate and the actual measurement of the weather station is 0.99, and the ECMWF is 0.89. Moreover, the time resolution in the previous studies (David et al., 2009; Alpert and Rubin, 2018) was also higher than 5 minutes. The link length used in these studies is 2-5 km, which is 4.8 km in this paper." (Page 3 line 70-75).

5.Lines 78 – 80: What kind of microwave link was used, i.e. commercial or for research, which brand, etc.?

**Response:** Thanks for your comment. We used a self-built test link, the brand is Siklu. We have added a relevant introduction to the paper and quoted the detailed information of this device as the following: "Fig. 1(a) shows the location and length of the test link and Fig. 1(b), (c) show the transmitter and receiver of the link. The link is 4.8 km long and operates at 73 and 83 GHz. We use Siklu's E-band radio transceivers (Siklu Carrier-Grade 1000Mbps E-Band radio, 2021) to transmit signals." (Page 3 line 87-90).

6.Lines 85–90: Could the authors provide a little more information about the weather station? On what elevation was it placed, also relative to the microwave link, was the measurement setup according to WMO standards, with what frequency is data collected? How representative are the measurements for the link path? The link path crosses a river, which may result in a slightly different humidity than what is measured at the weather station at the (seemingly) more urban site (but do correct me if I am wrong - I would like to see the authors' thoughts on this).

**Response:** Thank you for the comment. The weather station is placed on the ground below the weather tower, and one end of the link device is installed on the weather tower, about 29 meters above the ground. The data collection frequency of the weather station is 1 minute, which is set according to WMO standards. The installation site is in Xianghe County, but the link path does not cross any river. The link goes through the city. We have added the information in the paper required as explained above as the following: "We set up a two-way E-band millimeter-wave transmission link, one side of the link is installed on the top of a 29 m high meteorological tower, and the other side of the link is at the roof-top of a residential building." (Page 3 line 85-87) and "The weather station (German OTT Parsivel[2] Laser Raindrop Spectrometer, 2021) is placed on the ground below the weather tower. The data collection frequency is 1 minute, which is set according to WMO standards." (Page 4 line 96-98).

7.Lines 97–98 "median values": A question more out of interest, would it be helpful to also use other statistics of the received signal or even the full signal to get an idea of the uncertainty in the estimation process?

**Response:** Thank you for the comment. The advantage of the median is that it is not affected by large or small data. In many cases, it is more appropriate to use it to represent the general level of the overall data. And because it is necessary to determine an attenuation baseline for each drying period, and the calculation of the attenuation baseline also uses the median value, the analysis of the median value is indispensable.

8.Lines 107–108 "Since the quantization resolution of the equipment we have used is 1 dB and the quantification resolution of the water vapor density calculated by the weather station is 0.01 g/m, the resolution of the two data is inconsistent.": Could the authors elaborate a bit on this, also taking into account the length of the link path?

**Response:** Thank you for the comment. This is because the resolution of the data output by the link device is fixed. In fact, it has nothing to do with the path length of the link, it is determined by the system parameters of the link.

9.Lines 110–113: The moving average makes sense, I think. Have the authors, however, tested other moving window averages? I.e., where is the optimum and can we even go to higher temporal resolutions?

**Response:** Thank you for the comment. We tested different time windows and found that 60 minutes is the most appropriate. If the time window is lower than this value, the result after the moving average will not be smooth enough, and higher than this value will make the result after the moving average excessively smooth and distorted, and the hysteresis becomes obvious. Also, the time resolution after moving average is still 1 minute.

10.Lines 176–178 "We collected […] water vapor density.": This belongs in the methods section.

**Response:** We agree with the comment. We moved this sentence to the method section. (Page 10 line 191-193).

11.Lines 190–191 "There are many reasons for the error, and further analysis of the results is needed.": Can the authors elaborate in the discussion section on this? What errors are you thinking of and is there a threshold value for the cumulative attenuation value / water vapor density that makes it unfeasible for measurement?

**Response:** Thank you for the comment. One of the reasons for this error is: Since the measurement from the millimeter-wave link is a linear accumulation of integrated data, while the traditional weather station provides point measurement. In addition, the weather station in this paper is placed at one side of the link, so the estimated result will be different from the data of the weather station. There are also some environmental factors, such as the influence of wind speed, fog and dew on the antenna. We have added relevant explanations in the discussion section as follows: "Since the measurement from the millimeter-wave link is a linear accumulation of integrated data, while the traditional weather station provides point measurement. Therefore, the estimated result of the link estimation will be different from the data of the weather station. In addition, the weather station in this paper is placed at one side of the link and the observation from a weather station is only representative of the very local area where the equipment is sited which

may cause some deviations between the estimated results and the actual measured results. There are also some environmental factors, such as the influence of wind speed, fog and dew on the antenna." (Page 16 line 262-267).

12.Lines 197–205 / Eq. 15 – 17: This belongs in the methods section.
**Response:** We agree with the comment. We moved this sentence to the method section and added a new section on statistical tests. (Page 10-11 line 194-205).

13.Lines 209–211: Can I ask the authors to quantify the results in the text a bit more? Besides, what is 'good'? This needs a benchmark, so is it better than another model or estimation method?
**Response:** Thank you for the comment. We have provided more quantification for the results as follows: "It can be seen from Table 2 that the evaluation index also reflects that the the result of using millimeter-wave link to estimate the water vapor density is good. In other words, the highest monthly correlation is 0.95, the lowest root mean square error is 0.1, and the lowest mean relative error is 0.2." (Page 13 line 226-230). We compared the estimated results of the link with the prediction result of the European Centre for Medium-Range Weather Forecasts (ECMWF). The result is shown in Figure 6. It can be seen that our estimation result is closer to the actual measurement result of the weather station. In addition, we added this comparison result and evaluation index to the results section of the paper. (Page 15-16 line 249-260).

[Figure]

$PCC_{Link_{73}-WS}$ : 0.99
$PCC_{Link_{83}-WS}$ : 0.99
$PCC_{ECMWF-WS}$ : 0.82
$RMSE_{Link_{73}-WS}$ : 0.65
$RMSE_{Link_{83}-WS}$ : 0.58
$RMSE_{ECMWF-WS}$ : 4.79
$MRE_{Link_{73}-WS}$ : 0.06
$MRE_{Link_{83}-WS}$ : 0.05
$MRE_{ECMWF-WS}$ : 0.39

**Figure 6: The water vapor density graphs from the link, the ECMWF reanalysis and the weather station. These include the correlation, root mean square error and the mean relative error between the link and the weather station, the ECMWF and the weather station. WS = weather station.**

14.Lines 213–215 "Similarly, the evaluation result was better at 83 GHz in June, which shows that the 83 GHz millimeter-wave link has greater potential for water vapor inversion.": That makes sense, indeed, seeing Fig. 2 in the manuscript.
**Response:** Thank you for the comment. It can be seen in Figure 2 that the attenuation at 83 GHz is larger, which is more conducive to the extraction of attenuation caused by water vapor. Therefore, the estimation result of 83GHz is relatively better than that of 73GHz.

15. Lines 222–224 "Since the […] point measurement.": What do the authors mean with this sentence?

**Response:** Thank you for the comment. This sentence is to explain the reason for the error between the estimated value of water vapor density and the actual measurement mentioned in the paper. The observation from a weather station is only representative of the very local area where the equipment is sited. As the phrase "There are many reasons for the error, and further analysis of the results is needed." in the paper. (Page 11 line 220-221). We have added this sentence to the discussion section. (Page 16 line 262-263).

16. Lines 224 – 225 "In addition, the weather station in this paper is placed at one side of the link, so the estimated result will be different from the data of the weather station.": It may be worth mentioning this in either the methods or discussion section. Plus, what would this mean for the measurements?

**Response:** Thank you for the comment. The weather station and the link equipment work on different spatial scales, which may cause some deviations between the estimated results and the actual measured results. We have added relevant explanations to the discussion section as follows: "In addition, the weather station in this paper is placed at one side of the link and the observation from a weather station is only representative of the very local area where the equipment is sited which may cause some deviations between the estimated results and the actual measured results." (Page 16 line 264-266).

17. Lines 234–235 "vapor, so more accurate measurement means better response to the dangers facing humans and their environment (Fencl et al., 2020; Harel et al., 2015).": Very true. I think the authors can even make this stronger by mentioning the potential of new, opportunistic sensors here, which potentially gives a high(er) density of water vapor density sensors.

**Response:** We agree with the comment. We added this point to the paper as follows "Moreover, the millimeter-wave link also has the great potential to become a water vapor monitoring sensor, which can provide higher density water vapor data." (Page 17 line 291-292).

18. Line 236 "1 minute": But 60-min aggregations were used and validated, right?

**Response:** Thank you for the comment. We apologize for unclear expression. The time resolution of the link data we have used is 1 minute, and 60 minutes is just the window width for our moving average. Moreover, the time resolution of the data after the moving average is still 1 minute. We have added relevant explanations in the paper as follows "It is worth noting that the time resolution of the averaged data is still 1 minute." (Page 6 line 124).

19. Lines 236–238 "and used the […] within this year.": State here what the method was and especially what the authors did change, so what is new.

**Response:** Thank you for the comment. We apologize for unclear expression. We have added relevant explanations in the paper as follows: "and used the line-by-line

calculation of gaseous attenuation model provided by ITU-R to retrieve water vapor. We have proposed a new method which is to set a reference value for each dry period and give the upper and lower limits of water vapor attenuation. Then we applied this method to one year's data." (Page 17 line 294-296).

20.Lines 240–241: Besides the best reached values, also mention the average values to give the reader an idea of the overall model quality.

**Response:** We agree with this view. We've re-written the sentence in the manuscript as the following: "The highest Pearson correlation coefficient in summer months is 0.95, and the average value is 0.92. The lowest mean relative error is 0.05%, and the average value is 0.06%." (Page 17 line 298-299).

21.Lines 241 – 242 "Compared with previous studies, our water vapor inversion results have a higher time resolution.": How much higher?

**Response:** Thank you for the comment. The resolution of our estimation result is 1 minute, while the ECMWF is 1 day. Moreover, the time resolution in the previous studies (David et al., 2009; Alpert and Rubin, 2018) was also higher than 5 minutes. We have added relevant explanations in the paper as follows: "Compared with other studies, our water vapor inversion results have a higher time resolution, this is, our resolution is 1 minute, while the ECMWF is 1 day. Moreover, the time resolution in the previous studies (David et al., 2009; Alpert and Rubin, 2018; Fencl et al., 2020; Pu et al., 2021) was equal or greater than 5 minutes." (Page 17 line 300-303).

22.Lines 250–251 "This seasonal difference is also difficult to overcome. In the future, we will consider improving water vapor monitoring in winter in our research.": Could the authors share any thoughts on how to do this?

**Response:** Thank you for the comment. In this area of Hebei, the temperature changes with the seasons, and the temperature in winter is relatively lower than in other seasons. We consider that temperature can be used as an influencing factor to be linked to the change of water vapor density, thereby reducing the error of winter estimation.

23.Table 1: Could the authors add the number of minutes on record per indicated day? In addition, can the authors say something about the variation in the measurements during the days?

**Response:** Thank you for the comment. We have added the detailed number of minutes for each dry period in the paper (Page 5 line 104-106). There are 60 days of dry period, 59 days are 1440 minutes and 1 day is 1291 minutes. We think adding and repeating this explanation in the table is too lengthy, so we didn't add it in Table 1, instead we explained it in words.

24. Figure 3: the colors of the figure are not color-blind proof. I would recommend using different colors or to use a set of dashes in the lines. Besides, the figure is quite similar to Fig. 2 in Pu et al. (2021).

**Response:** Thank you for the comment. We have modified the Figure 3 to be color-blind proof (Page 9 line 166). In addition, this is a graph showing the changes in water vapor density at different frequencies for readers. Under the same temperature and pressure, the change of water vapor density is consistent, so Figure 2 will be the same as that of Pu et al. (2021). There are also some papers with similar graphs, such as Fencl et al. (2020) and Song et al.(2021).

25. Figure 4: What I miss in the results section (or accompanying discussion section) is a little discussion about any discrepancies we see between observations and model (e.g., but note that it is not limited to this, the differences we observe on the right of Fig. 4c and the drop in the simulations, but not the observations, in Fig. 4h). Can I ask the authors to comment on this? If the comments have a more speculative nature, this would very well fit in a discussion section. In any case, it could provide valuable information, also for the future development of such estimation methods.

**Response:** Thank you for the comment. This is the part worth discussing. We have added related some discussions in the discussion section as follows: "There are some unusual results in the measurement of the link. For example, from March 28 in Fig. 4 (a), the water vapor density estimated at 83 GHz is significantly higher than the estimated result at 73 GHz and the actual value measured by the weather station. This may be related to the extraction of attenuation. For this dry period, there is only one baseline that may not be able to accurately extract the attenuation caused by water vapor, and a more real-time reference value is needed. Although the calculation of the baseline may be one of the reasons for the error, this research has improved the real-time performance of the baseline and reduced errors compared with the studies of David et al., Alpert et al., and Fencl et al. (only set a constant baseline) (David et al., 2009; Alpert and Rubin, 2018; Fencl et al., 2020). Also, in Fig. 4 (c) and 4 (h), the estimated result of the water vapor density of the link decreases at the night of the day, which is caused by other uncertain sources. For example, changes in atmospheric refractive index and hardware-related artifacts may cause millimeter wave ray bending and RSL changes. This may also explain why the MRE in October in Fig. 5 (c) is higher than in other months." (Page 16 line 268-277).

26. Table 2 & Figure 5: The monthly overview is clear and interesting. Besides this overview, I think it may also be valuable to show a figure with at least some of the metrics per time step, to visualize the hour-to-hour or day-to-day discrepancies that can take place. That is valuable information for possible operational implementations of this method.

**Response:** Thank you for the comment. We have added the daily evaluation index of water vapor density inversion in Figure 5 (Page 15 line 244-245). The new Figure 5 is as follows:

[Figure]

**Figure 5: Daily and monthly evaluation index of water vapor density inversion(a) the correlation between the derived water vapor density and the reference, and (b) the root mean square error between the derived water vapor density and the reference. (c) the mean relative error between the derived water vapor density and the reference.**

27. Figure 5: What causes the high MRE peak in October, compared to the other months?

**Response:** Thank you for the comment. This is the part worth discussing. It is possible that the equipment is interfered by the external environment and the error of the system itself causes the attenuation value of the link to be higher in October, which leads to a large deviation between the inverted water vapor density and the measured value, that is, the MRE is higher. The dry period in October is 6 days in total. From the new Figure 5, it can be seen that the existence of one day of MRE is an extreme situation, which will

cause the monthly MRE to be high. We have added related some discussions in the discussion section (Page 16 line 276-277).

**Technical corrections:**

1. Line 30 - "The evaporation […]": The evaporation of water.

**Response:** Thank you for the comment. We have corrected this mistake and rewritten this sentence in the paper. (Page 1 line 30).

2. Lines 44–45 "the radiosonde is only launched about 2–4 times a day": Often even only once a day.

**Response:** Thank you for the comment. We have corrected this error in the paper (Page 2 line 45).

3. Line 100 "water vapor inversion": From a consistency perspective, would it better to use water vapor density instead of inversion?

**Response:** Thank you for the comment. What we want to express is that these data will be used in the inversion calculation of water vapor density. We have rewritten this sentence in the paper as follows: "These data will be used in the estimation of water vapor density." (Page 5 line 111).

4. Line 170 / Eq. 12: Is this a conditional statement? If yes, the notation should be slightly different (to make that clear).

**Response:** Thank you for the comment. This is a conditional statement. We have modified the notation of Eq. 12 as follows: "$RSL_i = \begin{cases} RSL_{low}, if\ RSL_i > RSL_{low} \\ RSL_{up}, if\ RSL_i \leq RSL_{up} \end{cases}$" (Page 10 line 187).

5. Line 179 "map": I think this should be 'graph'.

**Response:** Thank you for the comment. We have corrected this error and revised "map" to "graph" in the paper (Page 11 line 207-209).

6. Line 205 "which shows that the use of millimeter-wave link signal attenuation can estimate the water vapor density very well.": I think this can be removed, as the rest of the sentence already explains enough.

**Response:** Thank you for the comment. We have deleted this sentence from the paper. (Page 13 line 225).

7. Line 239 "obtained": I think this should be 'estimated'.

**Response:** Thank you for the comment. We have revised "obtained" to "estimated" in the paper (Page 17 line 297).

8. Line 240 "PCC" and "MRE": Better to write that out here, so Pearson's correlation coefficient and mean relative error.

**Response:** Thank you for the comment. We have rewritten this sentence in the paper as follows: "The highest Pearson correlation coefficient in summer months is 0.95, and the average value is 0.92. The lowest mean relative error is 0.05%, and the average value is 0.06%. The monthly and seasonal evaluation indicators show good results." (Page 17 line 298-300).

9. Figure 4: Something seems to go wrong with the figure headings, 'Spring and Summer' are on a different page than the rest of the figure. It is possible to either make the figure smaller or to make two figures out of it (e.g. spring + summer and autumn + winter). Besides, for comparison purposes, it helps when the y-axes are the same throughout the figure.

**Response:** Thank you for the comment. We modified the format of the title in Figure 4 in the paper (Page 12-13 line 222). We have also tried to keep the range of the y-axis consistent, but this will make it difficult to see the changes in the smaller water vapor density. For example, the water vapor density graph for December 2020. Figure 1 keeps the original y-axis, and Figure 2 is the adjusted y-axis (consistent with the y-axis of the water vapor density graph in July and August 2021). We can see that the change in water vapor density in Figure 1 is more obvious than that in Figure 2, so we did not change the y-axis.

[Figure]

**Figure 1: the original y-axis.**

[Figure]

**Figure 2: the adjusted y-axis.**

**References**

Alpert, P., Rubin, Y.: First Daily Mapping of Surface Moisture from Cellular Network Data and Comparison with Both Observa-tions/ECMWF Product. Geophysical Research Letters. 45, 8619–8628. https://doi.org/10.1029/2018GL078661, 2018.

David, N., Alpert, P., Messer, H.: Technical Note: Novel method for water vapour monitoring using wireless communication networks measurements. Atmospheric Chemistry and Physics. 9(7), https://doi.org/2413–2418. 10.5194/acp-9-2413-2009, 2009.

German OTT Parsivel² Laser Raindrop Spectrometer, 2021: http://www.sinokeytec.com/h-pd-248.html (accessed on 22 November 2021).

Fencl, M., Dohnal, M., Valtr, P., Grabner, M., and Bareš, V.: Atmospheric observations with E-band microwave links – challenges and opportunities. Atmospheric Measurement Techniques. 13(12), 6559–6578. https://doi.org/10.5194/amt-13-6559-2020, 2020.

Siklu Carrier-Grade 1000Mbps E-Band radio Datasheet. 2021: https://go.siklu.com/eh-1200-series-datasheet-new-lp (accessed on 17 November 2021).

Song, K., Liu, X., Gao, T., Zhang, P.: Estimating Water Vapor Using Signals from Microwave Links below 25 GHz. Remote Sensing. 13, 1409. https://doi.org/10.3390/rs13081409, 2021.

---

## Author Comment (AC2)

Dear reviewer:

  Thank you for your precious comments and advice. Those comments are all valuable and very helpful for revising and improving our paper, as well as the important guiding significance to our research. We have studied comments carefully and have made correction which we hope meet with approval. Revised portion are marked in red in the paper. The main corrections in the paper and the responds to your comments are as follows:

**Specific comments:**

  1.For Figure 1, please add the name for the horizontal axis and vertical axis.
  **Response:** Thank you for the comment. We have added the name for the horizontal axis and vertical axes to Figure 1 (a) (Page 4 line 93), and the revised Figure 1 (a) is as follows:

[Figure]

  2.For Figure 2, what time is used, local Beijing time, or the Coordinated Universal Time (UTC)?
  **Response:** Thank you for the comment. We are using Beijing time, which is China Standard Time (CST). We have explained in the paper. (Page 7 line 131).

  3. Line 86, rewrite as "The data recorded by the local weather station include humidity, ....".
  **Response:** We agree with the comment. We have revised this sentence in the paper. (Page 4 line 98).

  4.Lines 64–65 For each season, as shown in Figure 4, I would suggest the authors to use the same scale for the months within that season, which will be convenient for the readers to compare.?

**Response:** Thank you for the comment. We have also tried to keep the range of the y-axis consistent, but this will make it difficult to see the changes in the smaller water vapor density. For example, the water vapor density graph for December 2020. Figure 1 keeps the original y-axis, and Figure 2 is the adjusted y-axis (consistent with the y-axis of the water vapor density graph in July and August 2021). We can see that the change in water vapor density in Figure 1 is more obvious than that in Figure 2, so we did not change the y-axis.

[Figure]

**Figure 1: the original y-axis.**

[Figure]

**Figure 2: the adjusted y-axis.**

---

## Author Comment (AC3)

Dear Yoav Rubin:

Thank you for your precious comments and advice. Those comments are all valuable and very helpful for revising and improving our paper, as well as the important guiding significance to our researches. The main responds to your comments are as follows:

**Major comments:**

1.- Section 2.2: It is not clear how did you choose the period for calibration. How and for how long do you choose the period for estimating the median before calculating the humidity? It seems that you choose the dry period of each season. If so, why did you choose this approach? have you tried different approaches/periods? I think there needs to be more information about the method you used for retrieving the humidity.

**Response:** Thank you for the comment. We use the reference value calculated from the attenuation value of the current drying period as the baseline for the next drying period. Because the change trends of the two adjacent drying periods are similar, the calculated baseline can extract the water vapor attenuation value more accurately and reduce the error. This is more accurate than calculating only one reference value as the baseline for all drying periods.

2.- In section 2.1 you mentioned a very interesting point regarding the differences between RSLm at different seasons. I think it is one of your main conclusion that you should pay more attention to it. What are the main causes for these inter-seasonal changes? When do you see the sharpest changes? This information can be very helpful in the future for operational purposes.

**Response:** We agree with the comment. There are many reasons for seasonal variation. For example, changes in atmospheric refractive index and hardware-related artifacts may cause millimeter wave ray bending and RSL changes. In addition, in Hebei, the humidity in summer is greater than that in winter, and water vapor has a greater impact on the link in summer. We have added a discussion section, in conjunction with Figure 5, to analyze the causes of seasonal changes in more detail. (Page 15 line 254-279)

3.*This part of section 2.1 is related to the ref values for calibration in section 2.2. This order can be confusing. Maybe you should talk about it after explaining the "Principles of Estimating Water Vapor"

**Response:** Thanks for your comment. Because the received signal level RSL, pressure, temperature and other related data need to be used when explaining the principle of water vapor inversion, it is necessary to introduce the source of the data first. Second, the inversion method we use is based on the relationship between link attenuation and water vapor density, so we think we need to show the reader the trend of RSL and water vapor density changes before introducing the method.

---

## Referee Report (RR1)

Review of revised manuscript: **Water vapor estimation based on one-year data of E-band millimeter-wave link in the northeast of China** by Siming Zheng et al.

Ruben Imhoff

Ruben.Imhoff@deltares.nl

February 9, 2022

**Summary evaluation of revised manuscript**

I would like to thank the authors for their amount of work put into the revised version of the manuscript. The authors have tried to incorporate all given suggestions, generally up to a satisfactory level. I am also happy to see the comparison with the ECMWF estimates, which gives the statements in this work some extra strength.

I still have some minor suggestions, but after implementation of these suggestions/corrections, the manuscript is ready for publication, in my opinion. See below for my suggestions.

Sincerely,

Ruben Imhoff

**General comments**

*Comparison with ECMWF data*

Thanks a lot for incorporating this, I think it really strengthens the manuscript. Last note that I can make about it, is that I think the authors forgot to mention this comparison in the methods (it is mentioned in the abstract, results and so on, but not mentioned as a testing method).

*Response to previous comment*

*"Lines 110–113: The moving average makes sense, I think. Have the authors, however, tested other moving window averages? I.e., where is the optimum and can we even go to higher temporal resolutions?*

*Response: Thank you for the comment. We tested different time windows and found that 60 minutes is the most appropriate. If the time window is lower than this value, the result after the moving average will not be smooth enough, and higher than this value will make the result after the moving average excessively smooth and distorted, and the hysteresis becomes obvious. Also, the time resolution after moving average is still 1minute."*

Can I ask the authors to add their response (the one above) to the manuscript. I think it is valuable information to add to the methods sections where the 60-min window is introduced.

*Colorblind-proof figures*

Figures 4 and 6 are not colorblind proof. I would recommend the authors to make the figures colorblind proof (red - green is for instance a tricky one). Have a look at e.g. Crameri et al. (2020).

*Discussions section*

Suggestion: could the authors say something about the operational availability of the required CML data to scale their method up to an operational method in their study region and elsewhere? Thus, how likely is it that we will be using this method operationally in the near future?

**Specific comments**

Lines 25 – 26 "Compared with ECMWF reanalysis, the link performs better in water vapor density estimation": Quantify this a bit, so how much better (what are the correlation values of the tested estimation techniques)?

Lines 70 – 74: This is a result of the study. Although it is fine to mention why this study adds knowledge to existing literature (e.g. keep the mentioning of the higher temporal resolution and that you compared the results with among others ECMWF estimates), it is better to leave the results for the results section.

Lines 118 – 119 "Since the quantization resolution of the equipment we have used is 1 dB and the quantification resolution of the water vapor density calculated by the weather station is 0.01 $g/m^3$, the resolution of the two data is inconsistent.": This sentence still requires some extra information, e.g. why is the resolution of both data sets inconsistent?

Lines 225 – 228: Besides the good results that have been reached, also discuss the mean values and the moments when the estimation is not as good. That won't make the story weaker (not at all, actually), but gives an honest overview of the results.

Lines 255 – 260: also add the expected effect this has on the presented results.

**Technical corrections**

Line 25 – RMSE and relative error values: This is still missing the unit.

Line 40 – "measurement": measurements

Line 90 – "transmit": transmission

Line 105 – "60 dry periods with a duration of 1440 minutes": 60 dry periods with a duration of 1440 minutes per period.

Line 215 – "(Climate Overview of Hebei Province)": Is this a reference? If so, the year is missing.

Line 244 – "Fig. 6 show": shows.

Line 247 – "averaged daily": averaged per day.

Line 256 – "point measurement": point measurements.

Line 275 – "better than ECMWF": which product of ECMWF?

Line 285 – "high density": high spatial or temporal density?

Line 294 – "while the ECMWF is 1 day": while this 1 day for the tested ECMWF product.

Line 306: How do you propose to do that?

Figure 1: on the x-axis of sub figure (a) the degrees N are mentioned. This should be degrees E.

Figure 5 and 6: Something seems to go wrong with the dates on the x-axis, the spacing is no longer uniform. Besides, don't forget to add the units of the RMSE and MRE to the skill scores in figure 6.

**References**

Crameri, F., Shephard, G. E., and Heron, P. J.: The misuse of colour in science communication, Nat. Commun., 11, 5444, https://doi.org/10.1038/s41467-020-19160-7, 2020.

---

## Author Response (AR2)

**Changes in the Revision**

(1) Some sentences are rewritten.

(2) Figure 1 (a), Figure 4, Figure 5 (c), Figure 6, are all replaced.

(3) Table 2 has been revised.

**Responses to the Reviewer1's Comments**

Thanks for the reviewer to provide very useful comments and suggestions,

and please see our responses in the following:

**General comments**

*Comparison with ECMWF data*

Thanks a lot for incorporating this, I think it really strengthens the manuscript. Last note that I can make about it, is that I think the authors forgot to mention this comparison in the methods (it is mentioned in the abstract, results and so on, but not mentioned as a testing method).

**Response:** Thank you for the comment. We have added this comparison in the Materials and methods section as follows: "To more comprehensively test the link's ability to invert water vapour density values, we compare the results with the ECMWF reanalysis (CMIP5 daily data on single levels, 2021). The data source is water vapour density converted from daily near-surface relative humidity (with a horizontal resolution of 0.125° x 0.125°) obtained from ECMWF." (Page 5 line 106–108).

*Response to previous comment*

"Lines 110–113: The moving average makes sense, I think. Have the authors, however, tested other moving window averages? I.e., where is the optimum and can we even go to higher temporal resolutions?

Response: Thank you for the comment. We tested different time windows and found that 60 minutes is the most appropriate. If the time window is lower than this value, the result after the moving average will not be smooth enough, and higher than this value will make the result after the moving average excessively smooth and distorted, and the hysteresis becomes obvious. Also, the time resolution after moving average is still 1minute."

Can I ask the authors to add their response (the one above) to the manuscript. I think it is valuable information to add to the methods sections where the 60-min window is introduced.

**Response:** Thank you for the comment. We have added this explanation to the manuscript (Page 6 line 124-126).

*Colorblind-proof figures*

Figures 4 and 6 are not colorblind proof. I would recommend the authors to make the figures colorblind proof (red - green is for instance a tricky one). Have a look at e.g. Crameri et al. (2020).

**Response:** Thank you for the comment. We have modified the Figure 4 and 6 to be color-blind proof (Page 11-12 line 228, Page 14 line 261).

*Discussions section*

Suggestion: could the authors say something about the operational availability of the required CML data to scale their method up to an operational method in their study region and elsewhere? Thus, how likely is it that we will be using this method operationally in the near future?

**Response:** Thank you for the comment. We have added relevant explanations to the Discussion section as follows: "The research results show that it is feasible to invert water vapor using millimeter-wave links, and this method can be extended to the monitoring of meteorological factors such as rain, snow, and fog. At the same time, it also provides a basis for atmospheric monitoring of commercial microwave links, which will help to promote applications in the field of meteorology in the future. This is a test link, but E-band links are expected to be widely used in 5G networks and smart city networking." (Page 15 line 286–290).

**Specific comments:**

1.Lines 25 – 26 "Compared with ECMWF reanalysis, the link performs better in water vapor density estimation": Quantify this a bit, so how much better (what are the correlation values of the tested estimation techniques)?

**Response:** Thank you for the comment. We've re-written the sentence in the manuscript as follows: "Compared with ECMWF reanalysis, the correlation of the daily water vapor density estimation of the link has increased by 0.17, the root mean square error has been reduced by 3.14 g/m$^3$, and the mean relative error has been reduced by 0.33 %." (Page 1 line 25–27).

2.Lines 70 – 74: This is a result of the study. Although it is fine to mention why this study adds knowledge to existing literature (e.g. keep the mentioning of the higher temporal resolution and that you compared the results with among others ECMWF estimates), it is better to leave the results for the results section.

**Response:** We agree with the comment. We've re-written the sentence in the manuscript as follows: "Finally, a comparison between the link inversion results and the ECMWF reanalysis is given. The resolution of the link estimation result is 1 minute, while the ECMWF is 1 day. Moreover, the time resolution in the previous studies (David et al., 2009; Alpert and Rubin, 2018) was also higher than 5 minutes. The link length used in these studies is 2-5 km, which is 4.8 km in this paper." (Page 3 line 72–75).

3.Lines 118 – 119 "Since the quantization resolution of the equipment we have used is 1 dB and the quantification resolution of the water vapor density calculated by the weather station is 0.01 g/m3,the resolution of the two data is inconsistent.": This sentence still requires some extra information, e.g. why is the resolution of both data sets inconsistent?

**Response:** We agree with the comment. This is because the GUI of the wireless communication device cannot display the received signal level with higher accuracy, resulting in the link's estimated water vapor density value with a lower quantification

resolution than that calculated by the weather station. We've re-written the sentence in the manuscript. (Page 6 line 124-126).

4.Lines 225 – 228: Besides the good results that have been reached, also discuss the mean values and the moments when the estimation is not as good. That won't make the story weaker (not at all, actually), but gives an honest overview of the results.

**Response:** Thanks for your comment. We've re-written the sentence in the manuscript as follows: "In other words, the highest and lowest monthly correlation are 0.95 and 0.63, the highest and lowest root mean square error are 1.88 and 0.35, and the highest and lowest mean relative error are 0.27 and 0.05. June has the highest correlation and the root mean square error and mean relative error are also low. But January, February, and May 2021 have lower correlations." (Page 13 line 235-238).

5.Lines 255 – 260: also add the expected effect this has on the presented results.

**Response:** Thanks for your comment. We have added the expected effect this has on the presented results as the following: "This will reduce the accuracy of link inversion of water vapor density. As shown in Fig. 4, the results of the link inversion have the same trend as the measured values of the weather station, but they are not always consistent." (Page 15 line 270-272).

**Technical corrections:**

1.Line 25 – RMSE and relative error values: This is still missing the unit.

**Response:** Thank you for the comment. We have added units to the manuscript (Page 1 line 25).

2.Line 40 – "measurement": measurements

**Response:** Thank you for the comment. We have corrected this error in the manuscript (Page 2 line 42).

3.Line 90 – "transmit": transmission

**Response:** Thank you for the comment. We have corrected this error in the manuscript (Page 3 line 91).

4.Line 105 – "60 dry periods with a duration of 1440 minutes": 60 dry periods with a duration of 1440 minutes per period.

**Response:** Thank you for the comment. We have corrected this error in the manuscript (Page 5 line 110).

5.Line 215 – "(Climate Overview of Hebei Province)": Is this a reference? If so, the year is missing.

**Response:** Thank you for the comment. We have added year to the manuscript (Page 11 line 224).

6.Line 244 – "Fig. 6 show": shows.

**Response:** Thank you for the comment. We have corrected this error in the manuscript (Page 14 line 254).

7.Line 247 – "averaged daily": averaged per day.

**Response:** Thank you for the comment. We have corrected this error in the manuscript (Page 14 line 257).

8.Line 256 – "point measurement": point measurements.

**Response:** Thank you for the comment. We have corrected this error in the manuscript (Page 15 line 266).

9.Line 275 – "better than ECMWF": which product of ECMWF?

**Response:** Thank you for the comment. We have corrected this error in the manuscript (Page 15 line 291).

10.Line 285 – "high density": high spatial or temporal density?

**Response:** Thank you for the comment. We have corrected this error in the manuscript (Page 16 line 301).

11.Line 294 – "while the ECMWF is 1 day": while this 1 day for the tested ECMWF product.

**Response:** Thank you for the comment. We have corrected this error in the manuscript (Page 16 line 310).

12.Line 306: How do you propose to do that?

**Response:** Thank you for the comment. We have rewritten this sentence in the paper as follows: "We can try adding a temperature variable to the process of estimating the water vapor density." (Page 16 line 322).

13.Figure 1: on the x-axis of sub figure (a) the degrees N are mentioned. This should be degrees E.

**Response:** Thank you for the comment. We have corrected this error in figure 1 (Page 4 line 95).

14.Figure 5 and 6: Something seems to go wrong with the dates on the x-axis, the spacing is no longer uniform. Besides, don't forget to add the units of the RMSE and MRE to the skill scores in figure 6.

**Response:** Thank you for the comment. The smallest scale on the x-axis is the day, not the month. Each month does not contain the same number of days, so the month scale is not uniform. We have added units in figure 6 (Page 14 line 261).